# Developing a spatially explicit global oil and gas infrastructure database for characterizing methane emission sources at high resolution

Mark Omara[1,2], Ritesh Gautam[1,2], Madeleine A. O'Brien[2], Anthony Himmelberger[2], Alex Franco[1], Kelsey Meisenhelder[1], Grace Hauser[2], David R. Lyon[1], Apisada Chulakadaba[3], Christopher Chan Miller[3,4,5], Jonathan Franklin[3], Steve Wofsy[3], Steven P. Hamburg[1,2]

[1]Environmental Defense Fund, New York, NY, USA 10010
[2]MethaneSAT, LLC, Austin, TX, USA 78701
[3]School of Engineering and Applied Science/Department of Earth and Planetary Science, Harvard University, Cambridge, MA, USA 02138
[4]Center for Astrophysics, Harvard and Smithsonian, Cambridge, MA, USA 02138
[5]Climate Change Research Center, University of New South Wales, Sydney, New South Wales, Australia.

*Correspondence to*: Mark Omara (momara@edf.org), Ritesh Gautam (rgautam@edf.org)

## Abstract

Reducing oil and gas methane emissions is crucially important for limiting the rate of human-induced climate warming. As the capacity of multi-scale measurements of global oil and gas methane emissions have advanced in recent years, including the emerging ecosystem of satellite and airborne remote sensing platforms, a clear need for an openly accessible and regularly updated global inventory of oil and gas infrastructure has emerged as an important tool for characterizing and tracking methane emission sources. In this study, we develop a spatially explicit database of global oil and gas infrastructure, focusing on the acquisition, curation, and integration of public-domain geospatial datasets reported by official government sources, industry, academic, and other non-government entities. We focus on the major oil and gas facility types that are key sources of measured methane emissions, including production wells, offshore production platforms, natural gas compressor stations, processing facilities, liquefied natural gas facilities, crude oil refineries, and pipelines. The first version of this global geospatial database (Oil and Gas Infrastructure Mapping database, OGIM_v1) contains a total of ~six million features, including 2.6 million point locations of major oil and gas facility types and over 2.6 million kilometers of pipelines globally. For each facility record, we include key attributes—such as facility type, operational status, oil and gas production and capacity information, operator names, and installation dates—which enable detailed methane source assessment and attribution analytics. Using the OGIM database, we demonstrate facility-level source attribution for multiple airborne remote sensing detected methane point sources from the Permian Basin, which is the largest oil producing basin in the U.S. In addition to source attribution, we present other major applications of this oil and gas infrastructure database in relation to methane emission assessment, including the development of an improved bottom-up methane emission inventory at high resolution (1 km x 1 km). We also discuss the tracking of changes in basin-level oil and gas activity, and the development of policy-relevant analytics and insights for targeted methane mitigation. This work and the OGIM database, which we anticipate updating on a regular cadence, helps fill a crucial oil and gas geospatial data

need, in support of the assessment, attribution, and mitigation of global oil and gas methane emissions at high resolution. OGIM_v1 is publicly available at https://doi.org/10.5281/zenodo.7466757 (Omara et al. 2022)

## 1. Introduction

Limiting human-induced global warming, in accord with the climate-neutrality goals of the Paris Agreement (UNFCCC, 2015), requires "strong, rapid, and sustained" (IPCC, 2021) reductions in emissions of methane—a potent, but short-lived climate pollutant responsible for at least a quarter of today's gross climate warming (Myhre et al., 2013; Ocko et al., 2018; Ocko et al. 2021). Globally, the oil and gas sector accounts for about one-quarter of total anthropogenic methane emissions of around 360 teragrams (Tg) in 2017 (Jackson et al. 2021). By 2030, an estimated 50% of global oil and gas methane emissions have the potential for no-cost abatement relative to current emissions, because of the inherent commercial value of the recovered natural gas and widely available methane abatement technologies (Ocko et al. 2021). Recognizing the unique opportunity to slow the rate of near-term warming driven by avoidable methane emissions, a concerted effort toward fast, strategic action on emission reductions has emerged, with public commitments by oil and gas companies (OGCI, 2021) and pledges by countries (GMP, 2021) towards methane reduction targets and initiatives achievable within the decade.

At the same time, recent technological advancements in oil and gas methane emission quantification, characterized by a growing suite of airborne and satellite remote sensing instruments, have paved the way for rapid, frequent, and high-resolution mapping of both high-emitting methane point sources and area sources on a global scale (Jacob et al. 2022). These advancements in methane satellite remote sensing allow for the assessment of the temporal evolution of oil and gas methane emissions at multiple spatial scales, and enable the tracking of progress toward global emission reductions against stated mitigation targets. However, it is very challenging for satellite/airborne remote sensing to resolve facility-level attributes (such as oil and gas facility type or throughput rates) of detected methane sources, which must be paired with geolocated methane source datasets in support of source attribution, which is crucial for methane emissions monitoring and mitigation. Furthermore, methane emission rate estimations based on Bayesian inversion of satellite observations require a comprehensive, spatially explicit inventory of methane emissions as *a priori* information (Jacob et al. 2016), which invariably comes from bottom-up methane emission inventories dependent on geolocated oil and gas activity data (Scarpelli et al., 2022). Such geolocated methane source datasets must be global in scope, contain relevant attributional information on key oil and gas infrastructure types—including exploration/production, processing, refining, storage, and transmission facilities—that are important methane sources (EPA 2022, Alvarez et al. 2018), and can be updated on a regular cadence to account for evolving oil and gas activity.

The current dearth of an openly accessible, regularly updated, and global geospatial database of oil and gas infrastructure is a major limitation for methane source assessment and attribution of remotely-sensed emissions. There have been some useful efforts in the past to develop such a database. However, those were either limited in geographic scope, focused on one or a few oil and gas infrastructure types, or lacked granularity and regular updates (Carranza et al. 2018, Rafiq et al. 2020, Rose et al. 2018, GEO, 2018). In this study, we focus on the acquisition of public-domain location-specific datasets for all major oil and gas infrastructure types globally, including production wells, offshore platforms, natural gas compressor stations, processing facilities, liquefied natural gas (LNG) facilities, crude oil

refineries, and pipelines. The resultant geospatial database, which we refer to as the Oil and Gas Infrastructure
Mapping (OGIM) database (Omara et al. 2022), contains both locational information and, where available, facility-level attributes (e.g., facility type, operational status, and capacity or throughput) that is critical for methane source assessment and attribution.

## 2. Methods

### 2.1. Overview of global oil and gas infrastructure

Global oil and gas infrastructure is diverse, complex, and vast. Across global oil and gas producing fields or basins, oil and gas infrastructure plays a critical role in the extraction of oil and gas resources from underground reservoirs, as well as in gathering, treatment, compression, processing, refining, storage, and transportation of raw and refined products (Devold, 2013).

Oil and gas infrastructure in upstream operations enable the exploration, production, and gathering and treatment of oil and gas in both onshore and offshore locations. The major oil and gas facility types in upstream operations include (i) production wells, (ii) offshore platforms, and (iii) equipment or facilities that support oil and gas gathering, separation, metering, storage, and transportation. The latter may be collocated with well sites or operate as standalone facilities. Facilities in midstream operations allow for separation and treatment of raw natural gas to produce pipeline-quality dry natural gas and associated hydrocarbon products (Devold, 2013). These facilities typically include natural gas processing plants, natural gas compression facilities, LNG production (liquefaction) or regasification facilities, and gathering and transmission pipelines. The major facilities in downstream operations include crude oil refineries. Figure 1 shows examples of these major oil and gas facility types.

### 2.2. Open oil and gas geospatial data acquisition, integration, and database creation

We designed a three-step process for database development that involved acquisition of open geospatial data, data processing, and database analytics (Figure 2). We searched the web for open geospatial data on oil and gas infrastructure, focusing on major facility types that are relevant sources of measured methane emissions in upstream, midstream, and downstream operations, as described above. We used both semi-automated and manual web search approaches, acquiring and cataloguing open geospatial datasets retrieved from both official government and non-government sources. Non-government data sources included open data from oil and gas company reports, non-profit research institutions, academic research works, and other open oil and gas data websites (Figure 2). Where necessary, we used automated website translation services (Google Translate) for non-English websites for which relevant open geospatial datasets on oil and gas infrastructure were available. In cataloguing acquired datasets, each unique data source was assigned a source reference ID, and metadata associated with the downloaded datasets were recorded in a dedicated data catalog spreadsheet, including the URL links, the original data owner names, data file formats, the date the data was published and last updated, the date we last accessed the data, and how frequently the data was updated. The acquired geospatial datasets included several geospatial data file formats, such as GeoJSONs (.geojson), shapefiles (.shp), geodatabases (.gdb), delimited files (.csv, .dsv, .xls, .xlsx, .txt), and MS Access (.mdb, .accd).

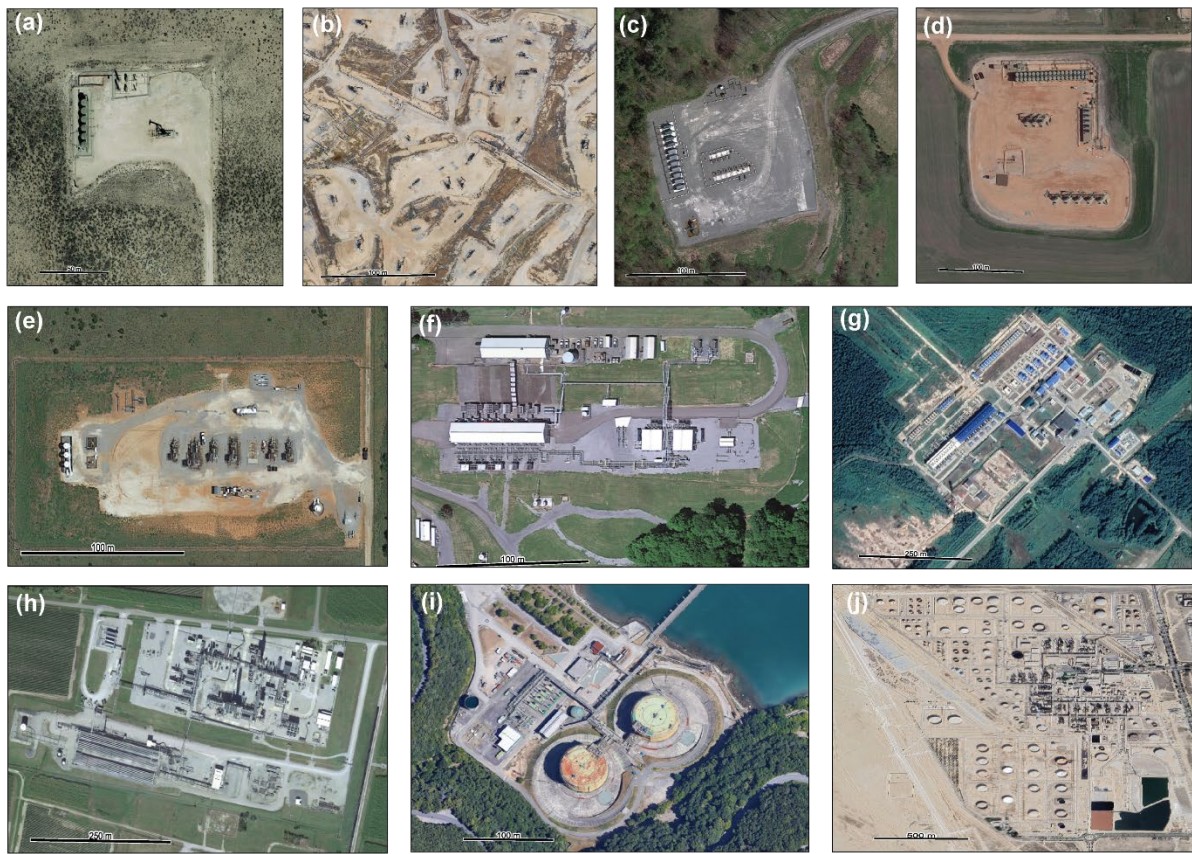

**Figure 1.** Examples of major oil and gas facility types as seen in high-resolution satellite imagery (Google basemap imagery, © Google Earth). (**a**). An oil well pad with one pump jack and storage tanks in the Permian Basin, U.S. (**b**). A cluster of oil pumpjacks in Kern River Oil Field in California, U.S. (**c**). A dry natural gas production well pad, with 11 horizontally-drilled wells, in the Marcellus Shale play (northeastern Pennsylvania, U.S.). (**d**). A mixed oil and gas production well pad in the Bakken Shale play (North Dakota, U.S.), with eight horizontally-drilled wells. (**e**). A natural gas gathering compressor station in the Anadarko Basin (Oklahoma, U.S.). (**f**). A natural gas transmission compressor station in Pennsylvania (U.S.). (**g**). A natural gas transmission compressor station in West Siberian region (Russia). (**h**). A natural gas processing plant in Louisiana (U.S.). (**i**). An LNG regasification facility in La Spezia, Italy. (**j**). A crude oil refinery in Mesaieed, Qatar.

For each country, we grouped all acquired geospatial datasets by their oil and gas facility categories (Table 1), which formed the basis for each of the geospatial data layers in the consolidated database. These facility categories relate to the previously defined major oil and gas infrastructure types in upstream, midstream, and downstream operations. In addition, where data were available, we included spatial information on major equipment (e.g., dehydrators at natural gas compressor stations) and components (e.g., valves at natural gas processing facilities). We also included, as its own data layer, locations of natural gas flaring at facilities or clusters of facilities, based on VIIRS (Visible Infrared Imaging Radiometer Suite) detections and gas flare radiant heat and gas flared volume estimates

from the Earth Observation Group (Elvidge et al. (2015)). Finally, where available, we included geospatial data for oil and gas fields, shale plays, and sedimentary basins.

We analyzed each acquired geospatial dataset, first by performing general data cleaning, such as identifying relevant data attributes (Table 2), replacing abbreviated attributes (e.g., for facility status) with full descriptions based on dataset metadata and/or documentation, and standardizing mixed data types. Further data pre-processing steps included standardizing data spatial references to an unprojected geographic coordinate reference system (datum: WGS 1984, European Petroleum Survey Group code, EPSG: 4326), automated translation of non-English attributes, and

standardization of date formats (OGIM format: "YYYY-MM-DD") and numeric fields (3 significant figures for all numeric fields, except latitude and longitude attributes, which we standardized to 5 decimal places). In addition, where data were available, we transformed all production, capacity, and throughput quantities to standard units of bbl (barrels) and Mcf (1,000 cubic feet) for oil and natural gas products, respectively. Similarly, where necessary, we converted pipeline lengths and field/basin areas to common units of kilometers (km) and square kilometers (km$^2$),

respectively.

In addition to the data cleaning and feature attribute standardization described above, additional data quality assurance and control that we applied before data integration included: (i) assigning standard missing data identifiers ("N/A" for categorical attributes, -999 for numerical attributes, and "1900-01-01" for date attributes), (ii) assessing and removing duplicate records, specifically for compressor stations and gas processing plants in the United States for

which multiple datasets were acquired, where obvious duplication of facility locations (within 100 m) and attributes (common facility and operator names) were identified and (iii) verifying and correcting facility category definitions (e.g., ensuring that the category for oil and gas wells included only well locations as opposed to offshore rigs) and locational information (e.g., confirming point locations of facilities match the state/province or countries to which they are attributed). As part of the data pre-processing, for each major facility category in each country, state, or

province, we automatically retrieved, visualized, and reviewed a subset of randomly sampled facility locations ($n$ = 20 to 50) in high-resolution satellite imagery, which provided an initial assessment of the accuracy of the facility category designation and spatial accuracy of point locations.

For some countries with rich open datasets on oil and gas infrastructure, especially Canada, it was necessary to merge multiple datasets for the same facility category to enhance the attributes integrated in the OGIM database.

For example, while we use surface hole locations for wells in the well location data for Alberta as integrated in OGIM database, we also incorporated information on well status, well name, operator name, and Universal Well Identifier (UWI) number found in a separate bottom-holes dataset from the Alberta Energy Regulator. When we merged multiple datasets from different sources to enhance the attributes for the same facility category in the same country, we identify these multiple data sources based on the source reference IDs as indicated in the data catalog.

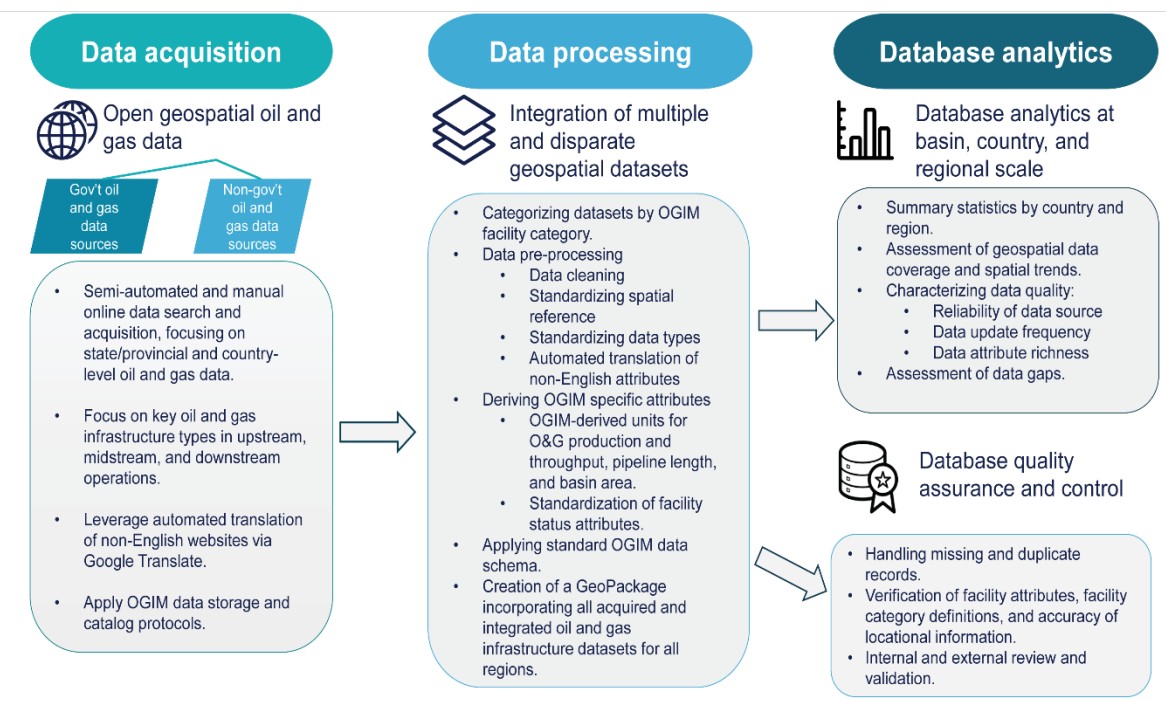


**Figure 2.** Data acquisition, data processing, analytics, and quality assurance and control procedures for the OGIM database development.

Finally, we also reviewed all unique descriptions of facility status information and included a standardized

facility status attribute ("OGIM_STATUS") in addition to the original facility status description, to facilitate grouping of infrastructure with the same level of activity. For example, original well status values of "active", "gas producer", "oil producer", "producer", "operating", "pumping", and "flowing" were all mapped to an "OGIM_STATUS" value of "producing".

As part of the data integration process, we developed and applied a standard data schema for each oil and gas

facility category included in the OGIM database (Appendix A). These data schema codified the data types, the geometry types and coordinate reference systems, as well as the feature attributes included in the OGIM database for all acquired datasets (Table 1, Table 2). The included feature attributes allow for facility localization (region, country, state, latitude, longitude, etc), identification (unique well identifier, facility name, operator name, etc), and characterization (e.g., facility type, installation dates, facility status, production rate, pipeline length, etc; Table 2). For

each unique facility category, we geospatially merged all the integrated datasets and exported the results into a GeoPackage layer. The final GeoPackage (the Oil and Gas Infrastructure Mapping database, OGIM_v1.gpkg) represents a consolidated database of all acquired and integrated open geospatial oil and gas infrastructure data across all regions.


**Table 1.** OGIM geospatial data layers.

| OGIM geospatial data layer | Additional information | Geometry type |
|---|---|---|
| Oil and natural gas wells | Includes active, inactive, and plugged and abandoned oil and natural gas wells. | POINT |
| Natural gas compressor stations | Facilities for natural gas compression in the gathering, transmission, and distribution sector. | POINT |
| Gathering and processing facilities | Includes natural gas processing plants, natural gas dehydration and other treatment facilities, and oil gathering and processing facilities. | POINT |
| Tank battery | Can be collocated with well sites; typical equipment includes oil and natural gas separation equipment and an arrangement of storage tanks. | POINT |
| Offshore platforms | Oil and natural gas drilling, production, and processing platforms in offshore areas. | POINT |
| LNG facilities | Includes both liquefaction and regasification facilities. | POINT |
| Crude oil refineries | - | POINT |
| Petroleum terminals | Includes tank farms and petroleum bulk storage tanks and terminals. | POINT |
| Injection, disposal, and underground storage facilities | - | POINT |
| Stations - Other | Includes metering and regulating stations and POL (petroleum, oil, and lubricants) pumping stations. | POINT |
| Equipment and components | Includes point locations for dehydrators, separators, tanks, and valves. | POINT |
| Oil and natural gas production | Includes reported well-level, facility-level, and field-level oil and natural gas production, as reported for 2021. | POINT |
| Natural gas flaring detections | Based on VIIRS natural gas flaring detections in 2021. | POINT |
| Oil and natural gas pipelines | - | LINESTRING |
| Oil and natural gas fields | - | POLYGON |
| Oil and natural gas basins | - | POLYGON |


**Table 2.** Examples of feature attributes for each layer in the OGIM database.

| Location attributes | Facility identification | Facility characteristics |
|---|---|---|
| Region | Unique well identifier | Facility type |
| Country | Facility ID | Facility operational status |
| State/Province | Data source reference ID | Installation date |
| On/Offshore | OGIM ID | Wells: spud date |
| Latitude | Facility name | Wells: completion date |
| Longitude | Operator name | Wells: drill type |
| geometry | | Wells: Annual Oil (BBL) and gas (MCF) production (as a separate data layer) |
| | | Pipelines: pipeline diameter (mm) |
| | | Pipelines: pipeline length (km) |
| | | Pipelines: pipe material |
| | | Pipelines: commodity |
| | | Compressor stations, processing plants, LNG, and refineries: reported capacity and throughput rates (BPD and MMcfd) |
| | | Fields and basins: area ($km^2$) |

### 2.3. Oil and gas geospatial database analytics

For each feature in the OGIM database, we assigned country names based on the UN Member States database (UN, 2022), with country boundaries based on a combination of ESRI World Country boundaries (ESRI, 2022) and the World Exclusive Economic Zones boundaries (EEZ, 2019). For each country, we used the annual country-level oil and gas production and consumption data based on international data from the U.S. Energy Information Administration (EIA, 2022) for 2019, the latest year for which complete oil and gas production statistics are available. From this data, we identify the major oil and gas producing countries that account for the top 80% of global oil and gas production (i.e., combined oil and gas production in energy units of barrels of oil equivalent per day (boed)). To analyze geospatial data on a regional basis, we group countries into seven regions based on the International Energy Agency's energy regions of (i) Africa, (ii) Asia Pacific, (iii) Central and South America, (iv) Eurasia, (v) Europe, (vi) Middle East, and (vii) North America (IEA, 2022).

We adapt the procedure by Rose et al. (2018) and develop geospatial data quality metrics, accounting for the reliability of the original data source, frequency of data updates, and richness of data attributes. In characterizing data source reliability, we considered the type of data source (i.e., government versus non-government data sources) and additional indicators such as ease of access of open data, and evidence of regular data updates and/or maintenance. We then assigned a score of 1—5 to each data source, with 5 representing highly reliable data source (updated frequently, available meta data and documentation, and data portals are well maintained) and 1 representing least reliable data source. We assessed the frequency of data updates based on the reported data update cadence for each

acquired dataset. We assigned a score of 1—5 to each data source, where (i) a score of 5 represents datasets that are updated on a daily to monthly cadence; (ii) a score of 4 represents data of quarterly to annual update frequency; (iii) a score of 3 represents data that is irregularly updated and was last updated in the past two to three years; (iv) a score of 2 represents data that is irregularly updated and was last updated within the last three to five years, and (v) a score of 1 represents data that was last updated more than five years ago. We also characterize the richness of feature attributes for each point feature in the OGIM database, focusing on oil and gas wells and major facility types in the midstream and downstream sector (i.e., natural gas compressor stations, gathering and processing facilities, LNG facilities, and crude oil refineries). For each feature for midstream facilities and crude oil refineries, we assessed whether the following six attributes were available: facility name, operator name, facility status, facility type, installation date, and capacity or throughput information. For each oil and gas well feature, we assessed whether the following six attributes were available: facility name, operator name, facility status, facility type, installation date (at least one of spud date or completion date), and drill type. If any of the attributes were available for each record, we assigned a score of 1 for that attribute for that feature, so that an attribute-rich feature will have a maximum score of 6. Thus, the maximum total score for each feature, accounting for reliability of data source, frequency of data update, and richness of feature attributes, is 16. We use this maximum total score to compute a normalized aggregate data quality score (0 to 1) for each feature, as well as mean normalized aggregate data quality scores for each country and region.

Understanding spatial accuracy of geospatial oil and gas data is important for accurate methane source attribution. However, it is not feasible to manually verify the accuracy of spatial information for the millions of point locations in the OGIM database. Nevertheless, and specifically for oil and gas wells, we identified a select number of countries and oil and gas producing basins in the U.S. (Bakken, Fayetteville, Permian, Marcellus, Denver-Julesburg), Mexico (Sureste), Argentina (Neuquén), Libya (Illizi-Ghadames), Saudi Arabia (Rub al Khali), Germany (northwest Germany), and Australia (Bowen-Surat) for an assessment of the spatial accuracy of point locations in the OGIM database. For each basin or country, we drew a random sample of 250 to 500 point locations, and automatically retrieved high-resolution satellite imagery (via the Google Earth satellite basemap, henceforth Google basemap imagery) at each location. On each retrieved image, we computed and plotted several buffers of different radii, representing distances of 10 m, 20 m, 50 m, 75 m, 100 m, and 150 m, around each selected point location (Figure 3). We then semi-automatically labelled each selected point location, indicating whether it was directly on a facility footprint as seen in satellite imagery, or offset within $x$ m of an actual facility footprint in satellite imagery, where $x$ is determined by the outline of the buffer radii around the target location (Figure 3). In total, we assessed the spatial accuracy of a random sample of 2,935 well locations in the selected basins mentioned above. In addition, we followed a similar procedure to evaluate the spatial accuracy of a random sample of natural gas compressor stations ($n = 550$ for randomly selected locations in Canada, the U.S., Argentina, Mexico, Nigeria, and Russian Federation), processing facilities ($n = 245$ for randomly selected facilities in Argentina, Canada, Mexico, and the U.S.), and crude oil refineries ($n = 301$ for randomly selected facilities in Argentina, Australia, China, and the U.S.).

| Directly on fac. loc | Within-10m | Within-20m | Within-50m | Within-75m | Within-100m |
|---|---|---|---|---|---|
| Within-150m | No clear fac. footprint | Uncertain | submit | prev | next |

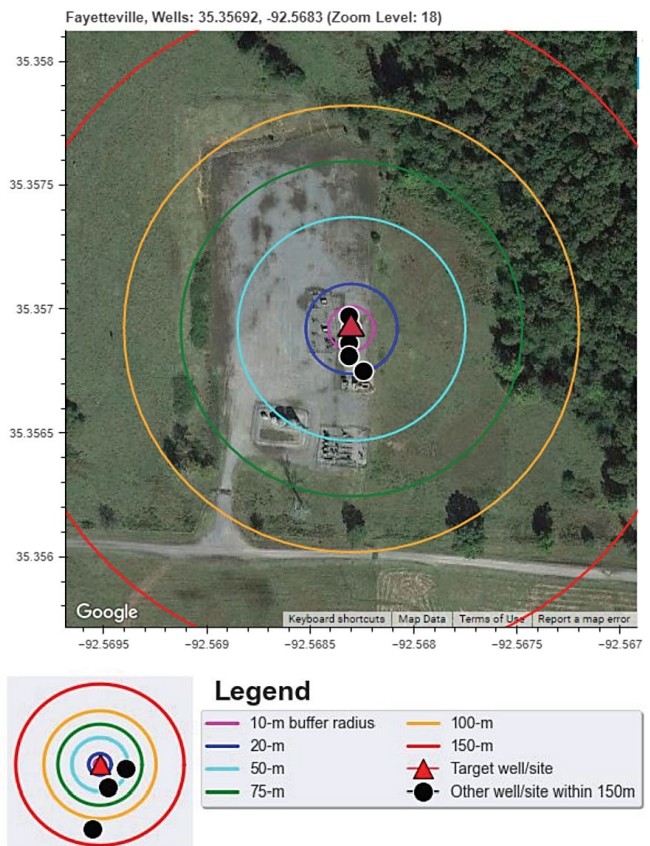

**Figure 3.** Schematic illustrating the semi-automated labelling of a random sample of oil and gas well locations in high-resolution Google basemap imagery (© Google Earth). The widgets are used to specify and automatically annotate the spatial accuracy of the randomly sampled facility location from the OGIM database (e.g., directly on facility footprint, within 10-m of the facility footprint, etc).

250

255

## 3. Results and discussion

### 3.1 Global overview

We acquired over 450 publicly available geospatial datasets of oil and gas infrastructure from 202 unique data sources. These datasets include all major oil and gas infrastructure types of interest (Table 1), although the total count and dataset availability for specific oil and gas infrastructure types exhibit wide variability among countries (Figure 4a). Among the major oil and gas producing countries representing the top 80% of global oil and gas production (EIA, 2022), publicly available government-sourced geospatial datasets accounted for two-thirds of the total, with countries in North America, South America (Brazil, Argentina), Norway, and Australia being notable for a large fraction of open-source government data in our consolidated database (Figure 4b). In contrast, nearly 80% of the acquired datasets for the bottom 20% of oil and gas producing countries came from non-government sources, reflecting a general paucity of reliable, open-source government-based oil and gas infrastructure datasets in these countries.

We acquired a total of ~six million geospatial data records, which includes point-based facility locations, oil and gas production, oil and natural pipelines, and fields and sedimentary basins. The vast majority of these records (~2.5 million records) are for oil and gas well locations, and roughly 85% of the records were sourced from countries in North America (Figure 5a). LNG facilities ($n$ = 338) and crude oil refineries ($n$ = 712) have the smallest representation in the database, although both show broad coverage globally (Figure 5a).

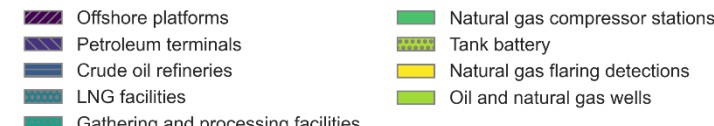

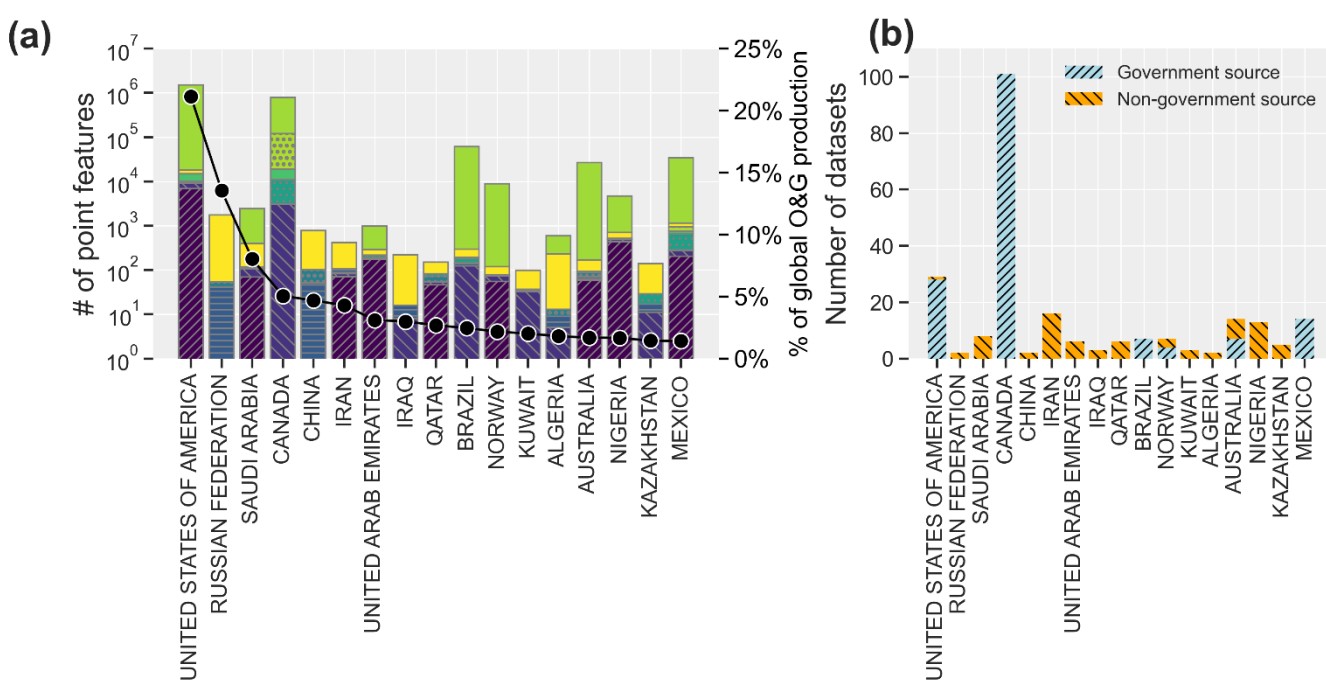

**Figure 4.** Summary statistics for acquired datasets for countries in the OGIM_v1 database. (**a**). Total count of acquired records for each oil and gas infrastructure type (bar plots) in each of the major producing countries that account for the top 80% of global oil and gas production. The right y-axis shows the percent contribution of country-level oil and gas production to global oil and gas production totals, based on EIA data for 2019 (EIA, 2022). (**b**). Total number of datasets and breakdown by government and non-government sources for the major producing countries that account for the top 80% of global oil and gas production. For these countries, government sources accounted for two-thirds (68%) of the total acquired datasets. For the remaining countries accounting for the bottom 20% of global oil and gas production, government sources made up roughly one-quarter (23%) of the acquired datasets.

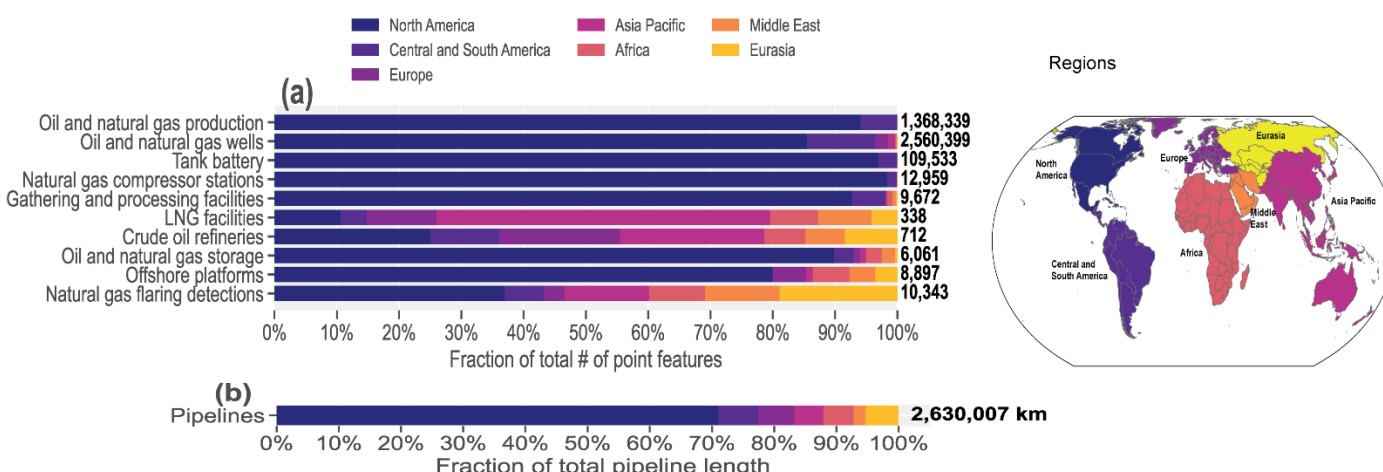

**Figure 5.** Summary statistics for major oil and gas facility categories by regions in the OGIM_v1 database. (**a**). Regional distribution of the total count of features for each oil and gas infrastructure type. The map shows the seven regions corresponding to the regions in (a). (**b**). Bar chart showing the regional distribution of the total pipeline lengths in the OGIM database.

In addition to point features, we acquired over 2.6 million km of oil and natural gas pipeline data globally, with a substantial proportion distributed in North and South American countries (Figure 5b).

### 3.2. Global spatial distribution of major oil and gas infrastructure data

The oil and natural gas wells data shows extensive coverage and significant overlap with major oil and natural gas basins for countries in North and South America, Europe (offshore regions), Australia, and New Zealand (Figure 6a). However, there is sparse open-source data availability for oil and natural gas well locations in several countries in Africa, Middle East, Eurasia, and parts of Asia-Pacific regions (Figure 6a). In addition, we find the largest density of open-source geospatial data for the major midstream oil and gas infrastructure, namely, natural gas compressor stations and oil and natural gas gathering and processing facilities, in countries in North and South America and parts of Eurasia (Figure 6b).

The natural gas flaring detections layer adds important spatial information on global upstream, midstream, and downstream natural gas flaring, revealing flaring hotspots in major oil and gas producing countries, including those for which limited open-source data on oil and natural gas well locations are available (Figure 6c, Figure 6a). With reported detections at over 10,000 facilities or facility clusters globally in 2021 (Figure 5a, Figure 6c), the available data allows for further methane source attribution as well as regional-scale emission characterization, as has been demonstrated in recent studies (Zhang et al. (2020), Lyon et al. (2021), Shen et al. (2022)). However, because the spatial resolution of the VIIRS instrument is ~750 m x 750 m at nadir (Elvidge et al. (2015)), linking VIIRS detections to individual oil and gas facilities presents certain challenges, particularly in oil and gas producing regions with spatially dense oil and gas facilities, such as in the Permian Basin in southern New Mexico and western Texas (USA). Such facility attribution requires further studies.

Globally, open-source data for oil and natural gas pipelines show broad coverage in North America, where spatial data for both gathering and transmission pipelines are available in several jurisdictions (Figure 6d). Outside of North America, the majority of acquired open-source data are for transmission oil and natural gas pipelines (Figure 6d).

We assessed the spatial density of all acquired oil and gas infrastructure datasets (excluding fields and basins), including natural gas flaring detections and oil and natural gas pipelines, at two spatial resolutions, namely, a regularly gridded, relatively granular spatial scale (25 km x 25 km, Figure 7a) and at the country scale (Figure 7b). On both spatial scales, we find the highest density of open-source geospatial records in North America, specifically the United States and Canada, with more than 2.5 million features (Figure 7b). In these countries, as well as in other countries with relatively high spatial density of open oil and gas infrastructure datasets (e.g., Mexico, Brazil, Argentina, Norway, Australia; Figure 7a, b), we can draw two broad conclusions: (i) most available datasets originate from authoritative government sources (Figure 4b), suggesting overall dataset reliability, and (ii) the highest density of oil and gas infrastructure locations are collocated with major oil and gas producing regions in these countries (e.g., Neuquén Basin in Argentina, Permian Basin in the United States, the North Sea; Figure 7a), suggesting broad open-source geospatial data coverage in support of comprehensive methane source attribution in such key oil and gas production basins.

Among the countries that account for the top 80% of global oil and gas production, countries with the lowest spatial densities of open-source geospatial oil and gas infrastructure data include the Middle Eastern countries (e.g., Saudi Arabia, Iraq, Iran, Qatar, Kuwait, United Arab Emirates), Algeria, Russian Federation, China, and Kazakhstan (Figure 7a, b). We note that these are also countries for which we acquired limited or no open spatial data on oil and gas well locations (Figure 6a), suggesting that the low spatial densities quantified in Figure 7 indeed reflect locational data gaps in our database, and not that such oil and gas infrastructure are absent in these countries. Equally of note is the predominance of centralized national oil company (NOC) operations, where public oil and gas data reporting policies vary widely, and general oil and gas data transparency have been previously described as "deficient" (Heller and Mihalyi, 2019). We discuss further below a quantitative assessment of the data gaps on a country-by-country basis, focusing on the major oil and gas producing countries accounting for the top 80% of global production.

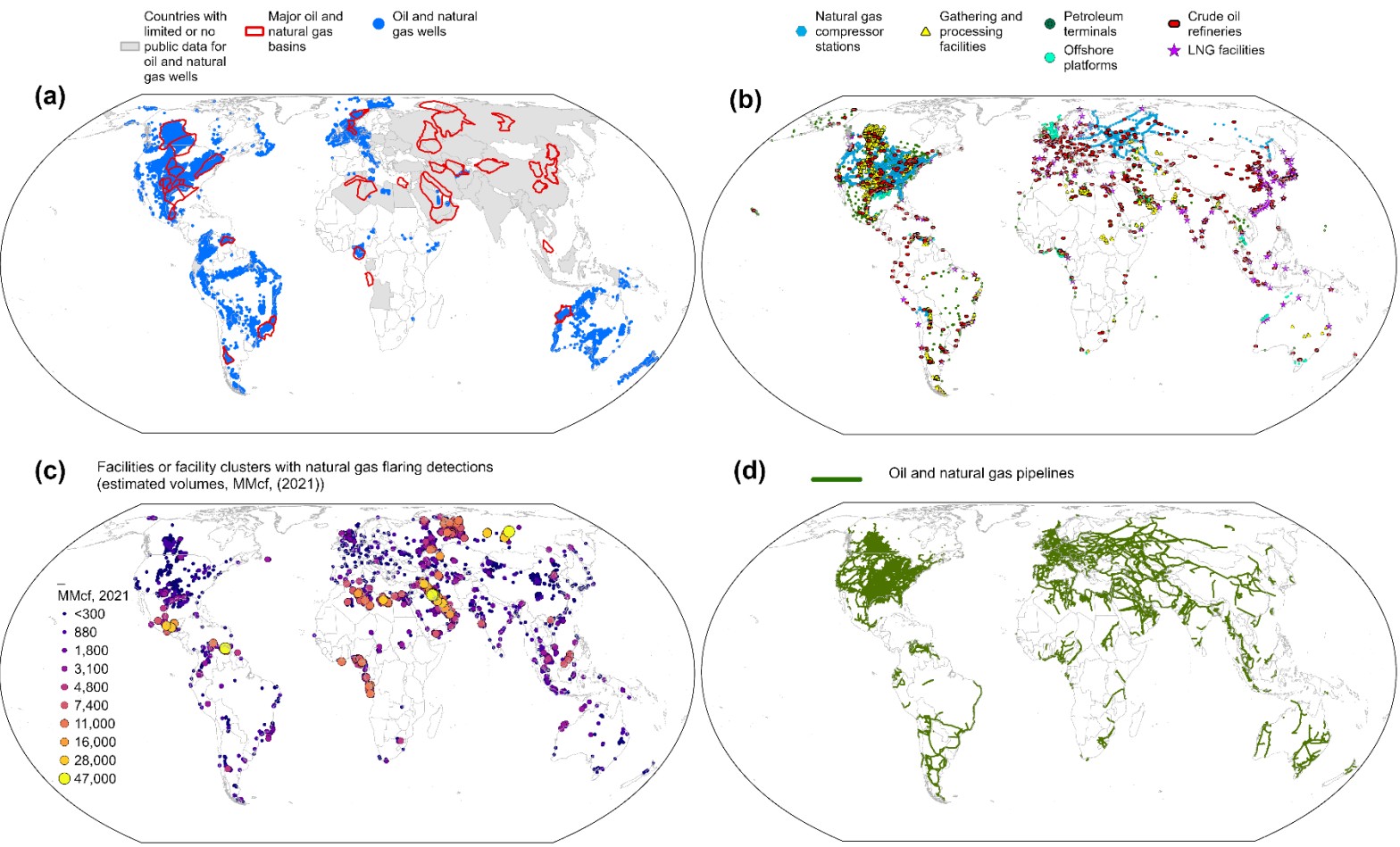

**Figure 6.** Spatial distribution of global oil and gas infrastructure locations in the OGIM_v1 database. **(a).** Spatial distribution of oil and natural gas wells, shown as blue points. The major oil and gas sedimentary basins accounting for the top 80% of global oil and gas production in 2019, are shown as red polygons. Countries for which there were limited or no public data acquired are shaded grey. **(b).** Spatial distribution of other major oil and natural gas infrastructure types, including natural gas compressor stations, gathering and processing facilities, petroleum terminals, offshore platforms, crude oil refineries, and LNG facilities. **(c).** Spatial distribution of natural gas flaring detections, based on VIIRS-derived datasets (EOG, 2022; Elvidge et al. 2015), highlighting global natural gas flaring hotspots based on estimated flared gas volumes. **(d).** Spatial distribution of acquired publicly available datasets for oil and natural gas pipelines.

### 3.3. Characterizing geospatial oil and gas data quality and spatial accuracy

The acquired datasets, which originate from 202 unique data sources, are expected to exhibit various levels of data accuracy, richness of data attributes, and frequency of data updates. We developed data quality scores for each feature in our database, quantifying richness of data attributes ("attribute score", range: 1-6), reliability of data source ("data source score", range: 1-5), and frequency of data updates ("update frequency score", range 1-5) to generate a normalized aggregate data quality score at the country level (range: 0-1; see Methods). We focus on country-scale

aggregate data quality metrics for oil and natural gas wells and for midstream infrastructure, specifically, natural gas compressor stations and oil and natural gas gathering and processing facilities. We quantitatively assess data quality in each country for which open oil and gas data for these facilities are available in the OGIM_v1 database.

Following a similar trend to the spatial data trends (Figure 7), we find the highest normalized aggregate data quality scores (>0.7) for countries in North America, South America (e.g., Brazil), Australia, and Europe (e.g.,

Norway; Figure 7). The defining characteristics of datasets in these countries that contribute to an overall high data quality score include: (i) data are sourced from transparent government sources, (ii) data is updated frequently (e.g., on a daily to monthly basis), and (iii) each feature include several key attributes such as facility name, activity status, facility operator, installation dates, and capacity or throughput information. In contrast, countries with low aggregate data quality scores < 0.5 (e.g., Russian Federation, Saudi Arabia, Iraq, Libya, Kazakhstan; Figure 8) are defined by a

general paucity of open geospatial oil and gas data from official government sources (Figure 4b), which is further compounded by infrequent data updates and limited attributes in available datasets from non-government sources. Even so, we note that each feature in OGIM_v1 is identified by its facility type which, in addition to location information, represents the minimum attributional information necessary for methane source identification (Cusworth et al. (2021), Irakulis-Loitxate et al. (2021)).

Given the importance of accurate locational information in facility-scale methane source attribution, we quantify the spatial accuracy of a subset of oil and gas facility locations in the OGIM database, based on semi-automated inspection of a random sample of 2,935 active oil and gas well locations in 11 major oil and gas producing basins against high resolution Google basemap imagery (see Methods). We find that, on average, 85% of this random sample of well locations had locational information that were accurate to within 20 m of actual oil and gas well pad

footprint as confirmed in high resolution satellite imagery (Figure 9), suggesting high spatial accuracy. Similarly, the percent of facilities that were located directly on the facility footprint or within 20 m of the actual facility footprint are 70%, 80%, and 83% for natural gas compressor stations, natural gas processing facilities, and crude oil refineries, respectively (Figure 9). The relatively low score for natural gas compressor stations is attributable to low location accuracy for facilities in western Canada, where facility locations are reported based on legal subdivision grids (e.g.,

the Dominion Land Survey grids, where each legal subdivision is ~400 m x 400 m). A small proportion of random samples (~2% to 5%) could not be quantified for spatial accuracy because no facility footprint was visible in satellite imagery, which may reflect the locations of recently constructed facilities assessed against an out-of-date satellite image.

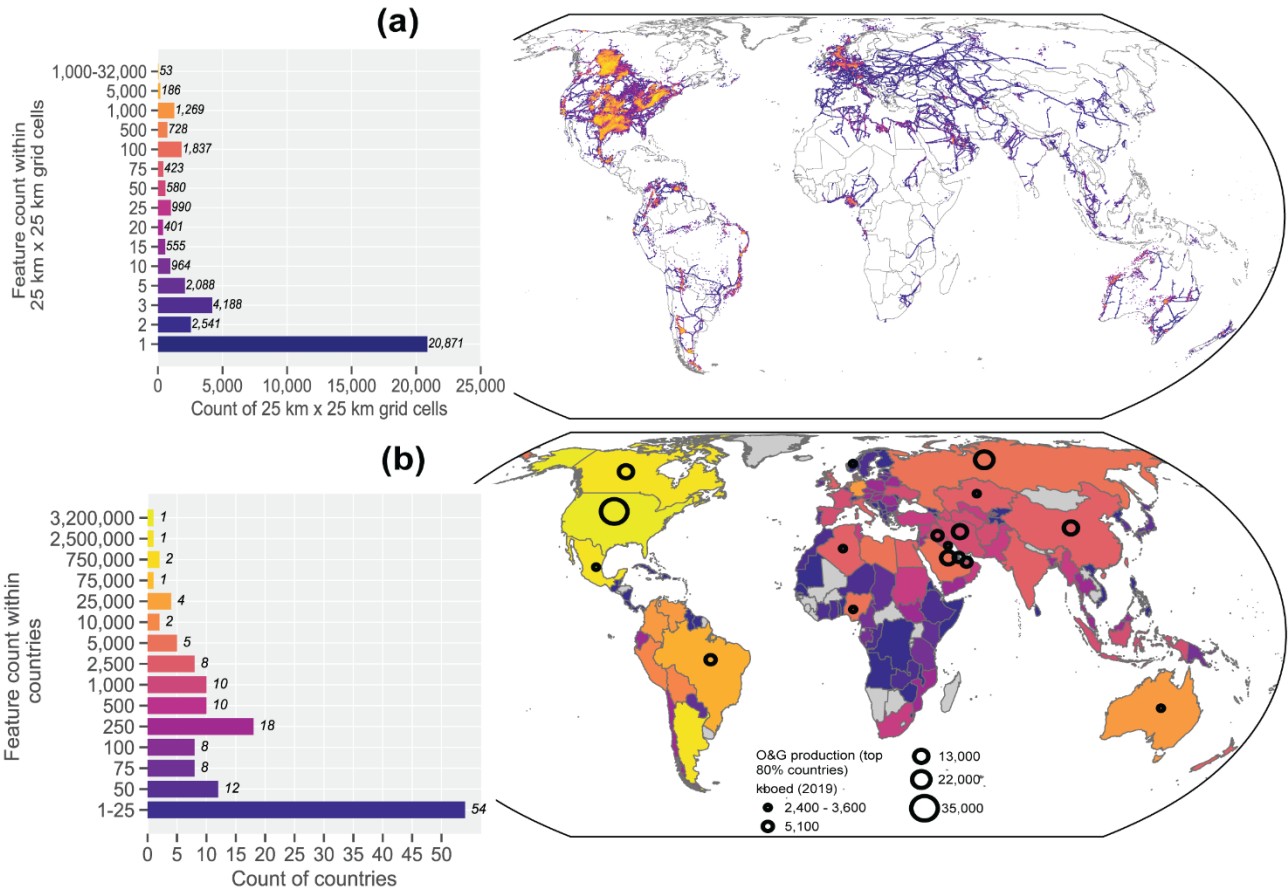


**Figure 7.** Spatial trends in acquired open oil and gas infrastructure datasets in the OGIM_v1 database. **(a).** Spatial densities of oil and gas infrastructure data on a regular grid of 25 km x 25 km. The bar chart represents the legend for the heatmap and shows the total count of features within each grid cell, ranging from 1 to 28,500. The *x* axis and the numbers on top of each bar show the frequency of grids cells with such feature counts; for example, there are 711 grid cells (25 km x 25 km each) that have between 1,000 to 28,500 features each. **(b).** Spatial densities of open oil and gas infrastructure data at the country scale. The bar chart represents the legend for the heatmap and shows the total count of features, ranging from 1 to over 3 million. The *x* axis and the value to the right of each bar shows the frequency of countries with such feature counts; for example, only one country (the United States) has over 3 million features. The graduated circles with black edges show the total oil and gas production for the countries that account for the top 80% of global oil and gas production (*n* = 17; EIA, 2022). Countries with no data are shown in grey.

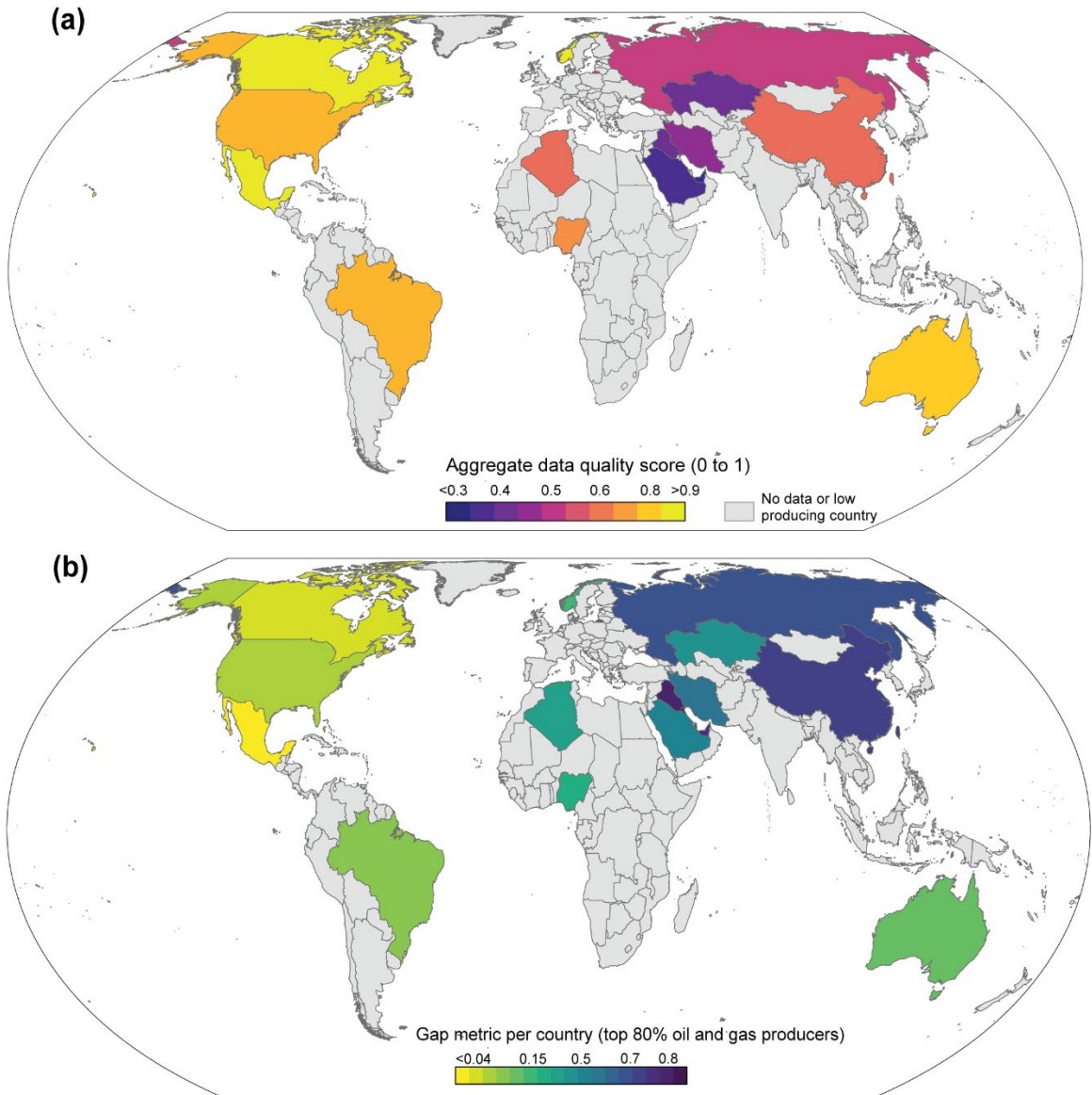

**Figure 8.** Characterizing geospatial oil and gas data quality and gap metrics. **(a).** The heatmap shows the normalized aggregate data quality scores at the country level, focusing on the top 80% oil and gas producing countries, and incorporating "attribute score", "data source score," and "update frequency score" (see Methods) for oil and gas wells and other major infrastructure types (i.e., natural gas compressor stations, oil and gas gathering and processing facilities, refineries, and LNG facilities). Spatial datasets of high data quality have high normalized aggregate data quality scores. **(b).** OGIM_v1 global data gap metric, focusing on the top 80% O&G producing countries. The gap metric aggregates the estimated data gaps for oil and gas wells, missing datasets for specific oil and gas infrastructure types, and gaps in data attributes, frequency of data updates, and reliability of data sources. Data-rich countries have very low scores less than 0.1, while data-limited countries have high data gap metrics greater than 0.5—0.7, on a scale of 0—1.

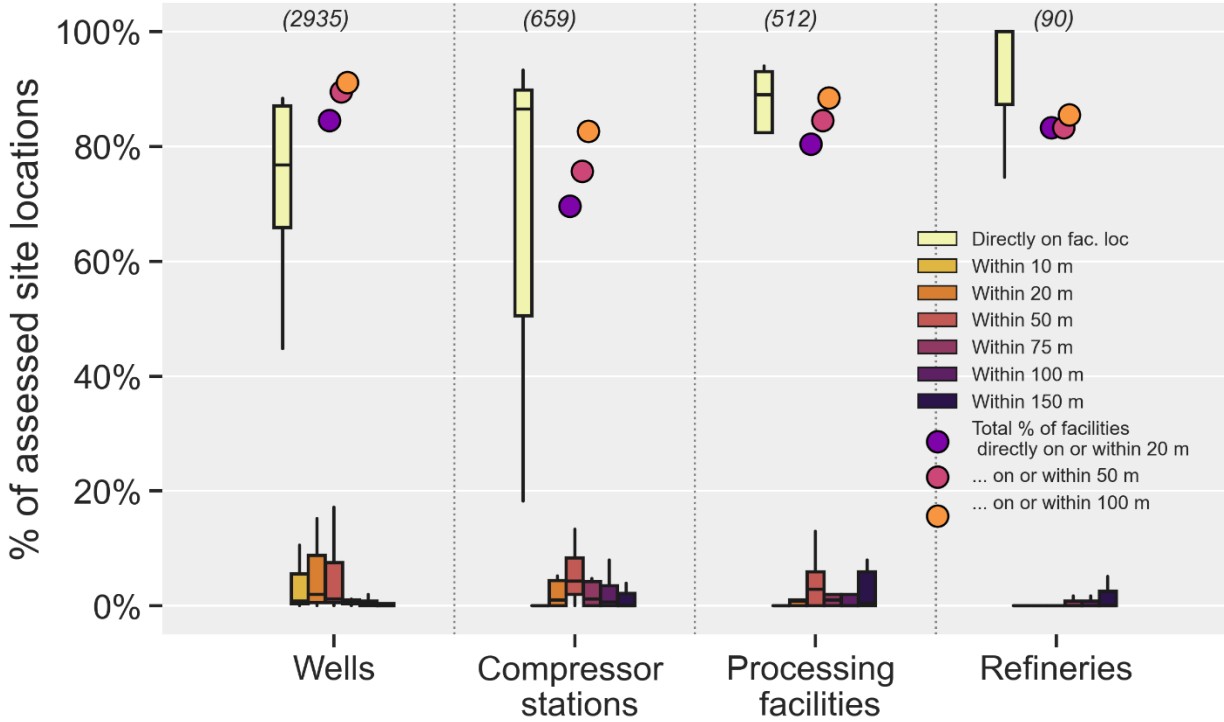

**Figure 9.** Characterizing spatial accuracy of oil and gas data. For each facility category, the box plots show the basin- or country-scale distribution of the % of labelled sites (random sample size shown in brackets in the top *x* axis) located directly on the facility footprint or offset within *x* m of actual facility footprint as seen in high-resolution satellite imagery. For example, for oil and gas wells, the first box plot indicates that among 11 basins, the median spatial accuracy is 76% for facility locations that were assessed to be located directly on facility footprint as seen in high-resolution imagery. The colored circles show the total percentage of all facilities with combined spatial accuracy of <20 m, <50 m, and <100 m. For example, for all 2,935 locations of wells, 85% are located directly on, or within 20-m of actual facility footprint as seen in high-resolution satellite imagery. A small fraction of facility locations (of roughly 2 to 5% of randomly sampled locations) were not visible in satellite imagery, likely because of outdated imagery (these are not shown in the figure). See supplemental information for additional assessment of the OGIM data coverage and spatial accuracy in the Permian Basin.

### 3.4. OGIM database gap assessment

Given our focus on acquiring public-domain datasets on oil and gas infrastructure, we acknowledge we are limited by open-access availability of geospatial datasets in regions and countries of interest. In general, we find wide availability of open-access oil and gas infrastructure datasets reported by governments in North America, parts of South America (especially Argentina and Brazil), parts of Europe (e.g., United Kingdom, Norway), and Australia. We quantify data gaps in the top 80% of oil and gas producing countries by assessing: (i) the expected number of oil and gas producing wells (the largest oil and gas infrastructure category in terms of total number of facilities), (ii) accounting for missing datasets for specific infrastructure categories, as well as (iii) the aggregate data quality score,

which incorporates existing gaps in the databases based on factors such as frequency of data updates and richness of data attributes.

We begin by analyzing basin-level oil and gas production data and assessing correlations with total number of producing wells and oil and gas productivity per well (barrels of oil equivalent per well) based on data for 52 basins in the Enverus Drillinginfo database (Enverus, 2021). We find a significant correlation between log-normalized oil and gas production (boe) and log-normalized well productivity (boe/well; $R = 0.48$, $p <0.0001$) and use this correlation to estimate the expected number of wells in each country in the top 80% of global oil and gas production based on EIA data (EIA, 2022). We assume data gaps exist if the estimated expected number of wells is more than the total feature count in the OGIM database. We then normalize the gap metric score from 0 to 1 such that a country with no publicly available data on wells gets a score of 0, while those countries with more well records than expected (as estimated above) get a score of 1.

In addition, we develop a presence/absence metric for the following seven major oil and gas infrastructure categories, including wells, compressor stations, processing facilities, refineries, LNG facilities, oil and gas offshore platforms, and storage facilities. For each country, we normalize this score from 0 to 1, such that any country for which the relevant data for all seven facility categories were acquired gets the highest score of 1. Finally, we incorporate the normalized aggregate data quality score (as previously discussed) to assess the overall data gap metric for each country, which we compute as the average of the three scores above.

Our gap assessment findings mirror the results of our aggregate data quality scores: we estimate few data gaps in oil and gas producing countries in North and South America compared to countries in North Africa, Middle East, Eurasia, and parts of Asia-Pacific where public-domain geospatial datasets on oil and gas infrastructure are of limited availability. Emerging approaches to fill in these data gaps involve the application of deep learning methods on high-resolution satellite imagery to automatically detect the locations of oil and gas infrastructure and classify by facility categories (Sheng et al. 2020). Further studies are needed to characterize the effectiveness of this approach for major oil and gas facilities, which are highly diverse in feature characteristics across global production regions.

## 4. OGIM database analytics

### 4.1. Tracking temporal changes in regional oil and gas activity

A global, open-source, and regularly updated database of oil and gas infrastructure with detailed attributes is important for understanding temporal changes in regional oil and gas activity, which in turn supports measurement-based characterization of regional methane emissions and emission trends. Where available, OGIM_v1 includes feature attributes such as spud and completion dates and oil and gas production for wells, which allows for tracking new well development and production trends (Figure 10a, b). For example, sustained growth in oil and gas production can be seen in the Marcellus (NE Pennsylvania, United States) and Neuquén Basin (Argentina), despite a general declining trend in the number of newly spudded wells between 2015 and 2021 (Figure 10a, b).

Natural gas flaring in oil and gas production, gathering, and processing has emerged in recent years as a crucial waste management issue with significant greenhouse gas, air quality, public health, and environmental justice implications (Zhang et al. 2019, Zhizhin et al. 2021, Plant et al. 2022, Blundell and Kokoza, 2022, Cushing et al.

2020). The open-source availability of quantitative data on the frequency of detected gas flaring and estimated flared gas volumes at global oil and gas infrastructure locations based on VIIRS remote sensing observations (Elvidge et al. 2015) and included in OGIM_v1 allows for the assessment of the temporal evolution of flaring activity in major production regions (Figure 6c) as well as progress toward global natural gas flaring reduction (World Bank, 2022; OGCI, 2021).

**4.2. Development of policy-relevant analytics and insights for methane emission assessment and mitigation**

The availability of open geospatial oil and gas production data in OGIM_v1 supports the characterization of measurement-based area-, regional- or national-scale methane loss rates (Alvarez et al. 2018, Zhang et al. 2020, Zavala-Araiza et al. 2021, Schneising et al. 2020, Omara et al. 2022, Shen et al. 2022) relative to production. The

480 assessment of measurement-based methane loss rate or intensity metrics is critically important for the evaluation of progress toward methane reduction targets such as the targets advanced by a consortium of major oil and gas companies (OGCI, 2021). For example, with a measurement-based methane loss rate of >3% of gross natural production (Zhang et al. 2020, Lyon et al. 2021, Chen et al. 2022, Shen et al. 2022), the Permian Basin in western Texas and southern New Mexico is one of the largest methane emitting oil and gas basins globally for which

substantial methane reductions are needed if methane intensity targets of <0.25% are to be achieved (OGCI, 2021).

While measurement-based facility-scale methane emission data or regional methane emission inventories are not included in the current version of the OGIM database, the available geospatial oil and gas infrastructure data can support the development of other policy-relevant analytics and insights that are crucial for targeted methane emission mitigation. The field- or basin-level characteristics regarding major oil and gas infrastructure (e.g., age of wells,

distribution of well-level production, and type and density of other major oil and gas infrastructure), oil and gas production profiles (e.g., oil-dominant, gas-dominant, or mixed oil and gas) and operational practices (e.g., asset consolidation by National Oil Companies or voluntary methane emission reduction measures put in place by specific operators) have the potential to influence the magnitude of measured methane emissions. As an example, Shen et al. (2022) reports extremely high methane loss rates of 13% relative to gross natural gas production in the Sureste oil and

gas production region of southern Mexico. The authors leverage detailed oil and gas activity data to postulate plausible mechanisms for high methane emissions observations, including (i) the potential venting of produced associated natural gas in a region with the largest density of newly drilled oil wells in Mexico, (ii) a concentration of central processing facilities previously identified with high potential for large methane emissions (Zavala-Araiza et al. 2021), and (iii) unique operational practices characterized by transportation and distribution of natural gas produced offshore

to onshore oil and gas infrastructure and partial gas utilization with potential for large releases. Such analytical insights, derived in part based on detailed oil and gas activity data, can help inform policy actions toward effective methane mitigation.

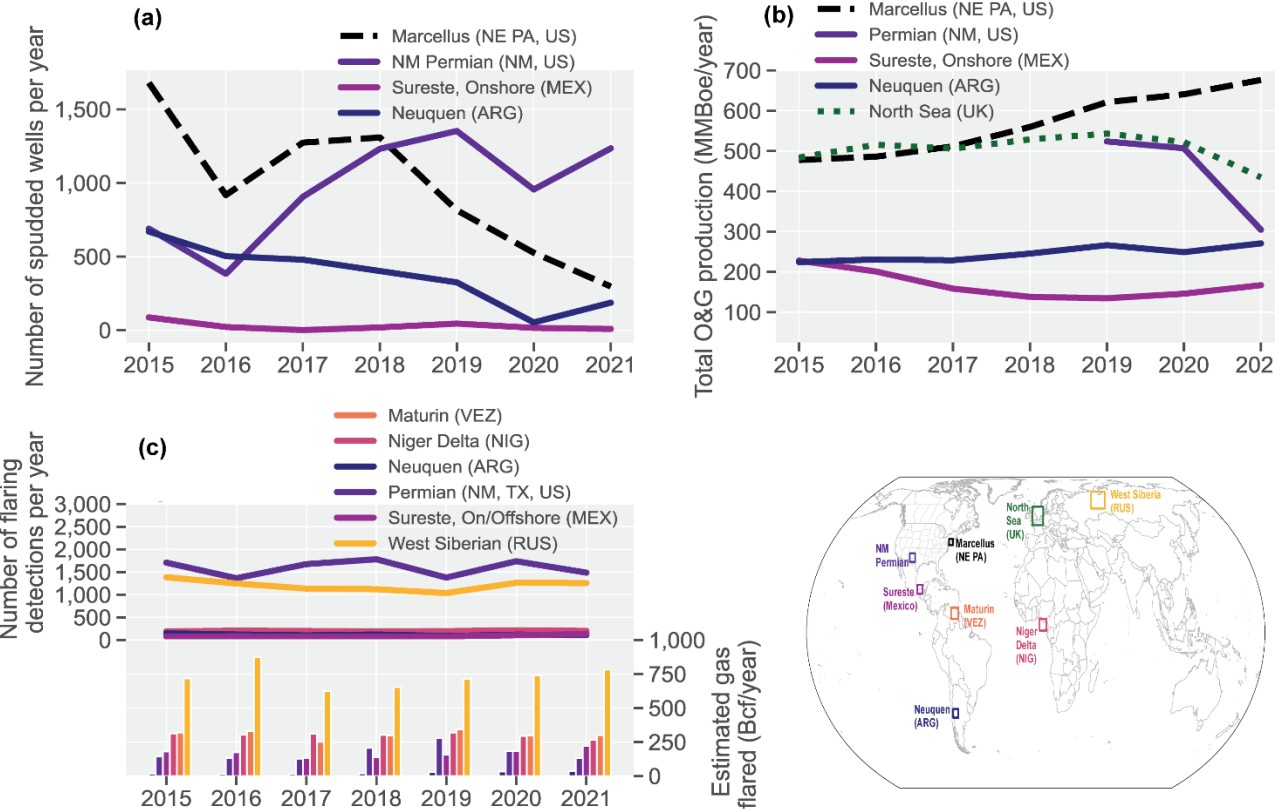

**Figure 10.** Temporal evolution of oil and gas activity indicators in select regions based on available data in OGIM_v1:
(**a**). Number of new wells spudded per year. (**b**). Total oil and gas production per year. (**c**). Natural gas flaring activity
per year. The top chart (left y-axis) shows the VIIRS-derived number of flaring detections per year, while the bottom
chart (right y-axis) shows the estimated annual gas flared volumes (based on Elvidge et al. 2015). The map shows the
approximate locations of select regions plotted in a-c.

To further illustrate this use case, we compare, in Figure 11, based on OGIM_v1 data, the distribution of well
age (based on reported spud dates as of 12/31/2021), well-level oil and gas production, and operator production
characteristics among three different oil and gas producing regions: (i) the New Mexico portion of the Permian Basin,
(ii) the Sureste region in southern Mexico, and (iii) the Neuquén Basin in Argentina. Among these regions, the number
of newly spudded wells per year (Figure 10a), well-level O&G productivity (Figure 11b), and the number of unique
operators and their oil and gas production (Figure 11c) are greater in the New Mexico portion of the Permian Basin,
even as the median age of active wells appear older than in the other two basins (Figure 11a). Furthermore, based on
total oil and gas production, the largest operator accounts for 10%, 60% and 90% of regional oil and gas production
in the NM Permian, Neuquén, and Sureste regions, respectively. These variabilities in regional density of oil and gas
infrastructure, production, and operational characteristics suggest that variabilities in underlying drivers of methane
emissions can be expected across various production regions, and that a tailored, rather than a one-size-fits-all strategy
for methane emissions mitigation may be required across international regions.

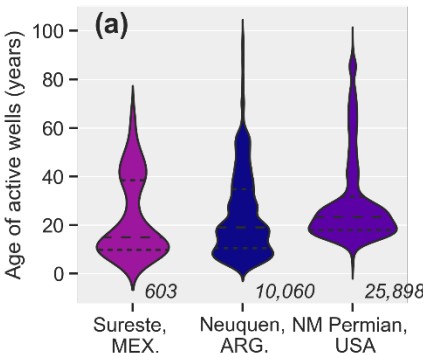 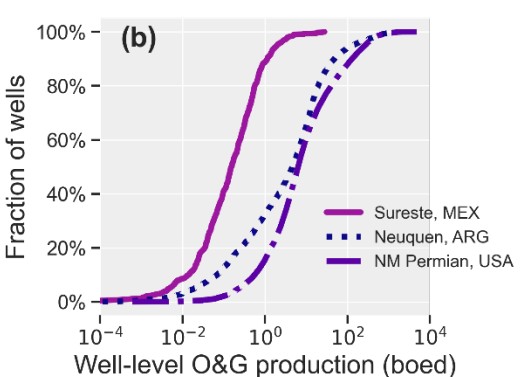 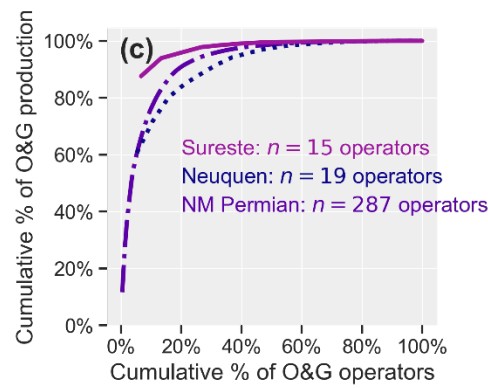

**Figure 11.** Examples of analytical insights derived from OGIM_v1 database. **(a).** Violin plots showing the distribution
of the age of active wells in the Sureste, Neuquén, and NM Permian (2021). The numbers at the bottom show the total
number of active wells in each region. **(b).** Cumulative distribution functions of well-level productivity, showing low
well-level productivity in the Sureste region (~90% of wells produce < 1 boed) and high well-level productivity in the
NM Permian (~10% of wells produce > 100 boed). **(c).** Lorenz curves showing the distribution of regional oil and gas
operators and their contribution to cumulative regional oil and gas production (e.g., the largest operator in the NM
Permian accounts for 10% of regional production).

### 4.3. Oil and gas methane point source attribution using airborne remote sensing

In recent years, the growing constellation, and advancement in capabilities of, methane remote sensing
satellites, have helped provide unprecedented insights into regional and point source methane emissions (Jacob et al.
2022). For example, recent research have underscored the importance of a small fraction of "super-emitting" and
"ultra-emitting" methane point sources as important contributors to global oil and gas methane emissions with
significant potential for cost-effective methane abatement (Varon et al., 2019, Cusworth et al. 2021, Cusworth et al.
2022, Irakulis-Loitxate et al. 2021, Lauvaux et al. 2022). Source attribution of these extreme methane point source
emitters have largely relied on facility type identification in high-resolution satellite imagery, with limited detailed
characterization of individual facility-scale sources. The available data attributes in the OGIM database—including
facility names, activity status, operator information, completion or installation dates, and production or throughput
data—support further source attribution analytics beyond facility type identification. Such source attribution analytics
have the potential to provide further key insights into the characteristics of high-emitting point sources across oil and
gas production regions. For example, by assessing the age of extreme methane emitters, Irakulis-Loitxate et al. (2022)
reported the detection of more extreme emissions from newer facilities <2 years old in the Permian Basin and
estimated that newer facilities contribute twice as much more methane than older facilities.

In Figure 12 and Table 3, we show five examples of detailed methane source attribution for high-emitting
point sources detected in the Permian Basin based on observations from an August 2021 deployment of MethaneAIR,
an airborne precursor mission for MethaneSAT (Staebell et al. 2021), which is an upcoming satellite mission managed

 by MethaneSAT LLC – a wholly owned subsidiary of Environmental Defense Fund. For each example of a high-emitting

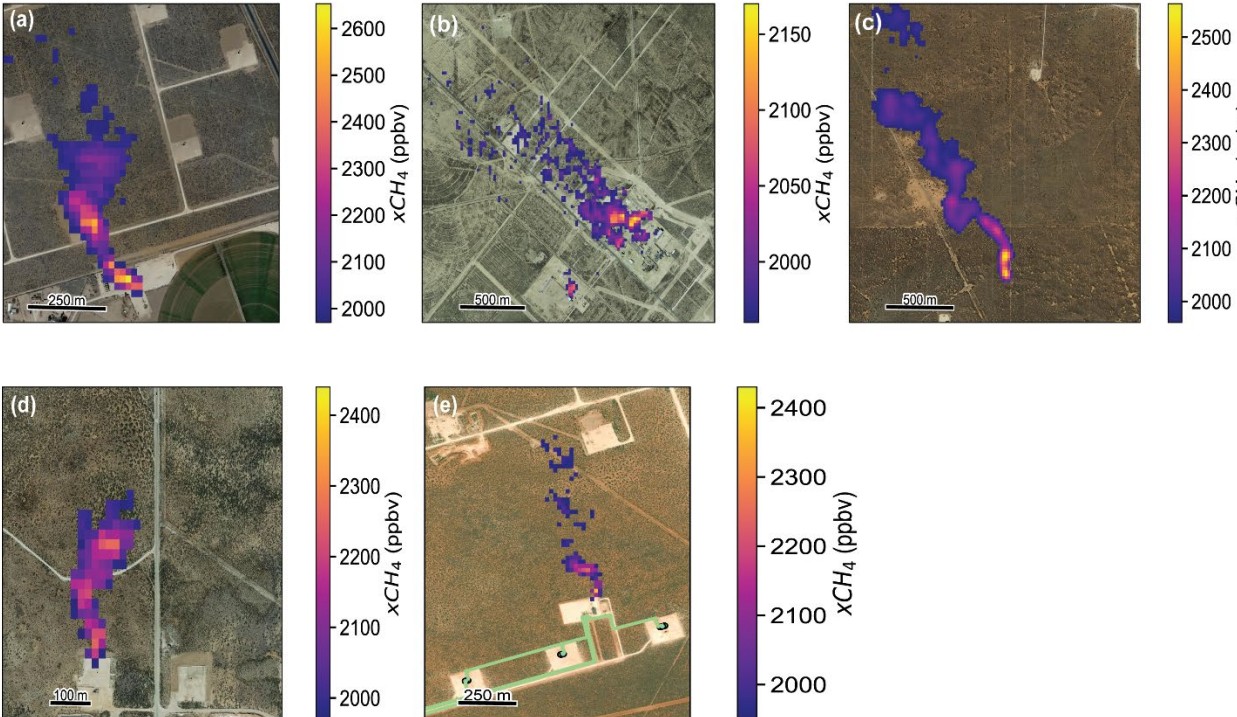

**Figure 12.** Detailed methane source attribution of high-emitting point sources based on OGIM_v1 data as visualized in high resolution Google basemap imagery (© Google Earth). The high-resolution $xCH_4$ data shown here are from an August 2021 deployment of MethaneAIR (an airborne precursor instrument for MethaneSAT), geo-rectified and regularly gridded on a 25 m x 25 m grid. Detailed source attribution for each plume is provided in Table 3. **(a).** Source identification of an oil well pad with four horizontally drilled oil wells. **(b).** Source identification of a 200 MMcfd natural gas processing plant. **(c).** Source identification of a methane high-emission event at a natural gas gathering pipeline segment. For this site, we also reviewed Sentinel-2 imagery at 10 meter spatial resolution (acquired in August 2021) and found no other major oil and gas infrastructure was located or was actively being developed in the area close to the plume origin. We supplement OGIM_v1 data with operator-reported emission incident report to the New Mexico Oil Conservation Division (OCD, 2021), which identified a major natural gas release due to a rupture at a weld along the pipeline segment at this location. The operator estimated a total natural gas release of 9,620 Mcf over a duration of 18 hours. Based on this information, we estimate a methane vent rate of 8.2 metric tons per hour at the time of observation, assuming 80% methane content in gathered natural gas. **(d).** Source identification of an oil well pad with one horizontally-drilled well in the Permian Basin. **(e).** Source identification of methane emissions from a central gathering facility servicing three well pads with five wells. The gathering pipelines connecting the wellheads to the facility are shown in green.

**Table 3.** Methane source attribution of high-emitting point sources in the Permian Basin (as shown in Figure12).

| Methane emitting oil and gas facility | (a) | (b) | (c) | (d) | (e) |
|---|---|---|---|---|---|
| Facility type | Oil well pad | Natural gas processing plant | Natural gas gathering pipeline | Oil well pad | Central gathering facility |
| Facility name | Haag Hz unit | Mi Vida Gas Plant | Lotus Lateral Poly 8' | Nash Unit | Rustler |
| Operator name | Earthstone | Energy Transfer | Lucid Energy | XTO | EOG |
| Facility status | Producing | Operational | Operational | Producing | Producing |
| Other facility attributes: | *Age:* 2 years | *Installed capacity:* 200 MMcfd | *Reported emissions event:* August 24, 2021 | *Facility age:* 9 years | *# of wells:* 5 |
| | *# of wells*: 4 | *# of compressor units:* 8 | *Reported vented gas*: 9,620 Mcf over 18 hours | *Gas production:* 160 Mcfd | *Drill type:* Horizontal |
| | *Drill type:* horizontal | *Facility age:* 7 years | | *Oil production:* 22 bpd | *Gas production:* 1,200 Mcfd |
| | *Gas production:* 1,000 Mcfd | | | # of wells: 1 | *Oil production*: 660 bpd |
| | *Oil production:* 550 bpd | | | *Drill type:* horizontal | |

point source, we query and retrieve key attributes from the OGIM_v1 database (e.g., facility age, operator, production, and throughput capacity) that further improve our understanding of the methane emitting source. In addition to location information, facility ownership attribution is possible, where such data are available, potentially enabling rapid abatement of detected extreme methane emissions when near-real time intelligence on high emissions is transmitted

to the known responsible operator. However, we note that several other factors can influence the ability for facility-scale methane source attribution, including the spatial resolution of methane plume detection, the density of oil and gas infrastructure, and co-location or lack thereof, with other non-oil and gas methane emitting sources within individual remotely-sensed methane footprint.

## 5. Improvements to bottom-up oil and gas methane emission inventories

Bottom-up oil and gas methane emission inventories are important for assessing regional and country-specific trends in methane emissions and form the basis for regulatory measures designed to mitigate methane emissions from key sources in oil and gas operations (EPA, 2022; UNFCC, 2022). Typical methods for the development of these inventories involve the application of methane emission factors (e.g., methane emitted per unit of activity) to activity data (e.g., total number of producing wells; EPA, 2022). The accuracy and completeness of these inventories are dependent, in part, on the representativeness of the methane emission factors and the comprehensiveness of the oil and gas activity data. As methane remote sensing has advanced in recent years, an important need for accurate, spatially representative, and high-resolution bottom-up methane inventories has emerged, since these inventories can function as *a priori* information required for the Bayesian inversion modelling framework typically used for methane flux rate quantification (Jacob et al. 2016).

The OGIM database supports improvements and updates to existing bottom-up methane emission inventory estimates by providing open-access spatially explicit data on facility locations and their attributes (activity data). We suggest that, where available, these detailed open access oil and gas infrastructure data and attributes can be integrated with empirical or modelled facility-scale methane emissions/emission distributions (based on measurements at representative sites) to update and improve current estimates of total oil and gas methane emissions, in addition to providing high-resolution gridded methane inventories needed for Bayesian inference of satellite observations. Below, we discuss the application of these principles to the development of gridded bottom-up oil and gas methane emissions inventory for the Permian Basin, which updates a previous inventory reported in Zhang et al. (2020) using similar methodologies described therein (and in the Supporting Information), but with updated activity data and site-level methane emissions characterization, modelling, and extrapolation to the full population of facilities in this region in 2021.

We begin by compiling oil and gas activity data in the Permian Basin, based on OGIM data, and supplementing, where needed, with proprietary data, particularly for well-level oil and gas production (Enverus, 2021), which is currently not publicly reported for the state of Texas. To estimate site-level methane emissions for oil and gas production well sites, we estimate the total number of actively producing well sites following the geospatial clustering approaches outlined in Omara et al. (2022). Table 4 shows the summary of the activity data for the Permian region for 2021, while Figure 13 shows the spatial distribution of oil and gas infrastructure, production, and gas flaring in this region.

We leverage existing site-level methane emission data and develop representative methane emission models to estimate total regional methane emissions, given the total population of operational oil and gas facilities in the region. Briefly, for oil and gas well sites, we use the site-level emissions data and the emission models developed by Omara et al. (2022) to estimate total methane emissions for low-producing well sites ($n = 104{,}100$), defined as well sites that produce less than 15 barrels of oil equivalent per day per site. For non-low production well sites ($n = 27{,}171$), we develop an emission factor of 3.6% (95% CI: 2.2—6.2%) methane loss rate relative to site-level methane production based on lognormal fit ($EF = exp(\mu + 0.5\sigma^2)$, where $\mu$ = -1.76 (-1.9, -1.5) and $\sigma$=2.4 (2.3—2.6)) to site-level methane loss rate measurements taken at 753 non-low production well sites and reported in previous studies

(Caulton et al. (2019), Brantley et al. (2014), Robertson et al. (2017, 2020), Omara et al. (2016, 2018)). To be conservatively low, we report our estimate of total well site emissions based on the lower bound of the modelled

methane emission loss rate of 2.2%. For natural gas gathering and boosting compressor stations, we generate site-average methane loss rates of 0.25% of natural gas gathered based on an updated national estimate of gathering compressor station emissions by Zimmerle et al. (2020). We assume an upper bound of +50% on the total number of natural gas gathering compressor stations in the Permian Basin, and model their methane emissions assuming a triangular distribution for station count, with mode and minimum set at 837 stations and maximum at 1,256 stations.

For natural gas processing plants and transmission compressor stations, we use the emissions distributions and emission factors as modelled by Alvarez et al. (2018) based on facility-level methane emission measurements from Mitchell et al. (2015), Marchese et al. (2015) and Zimmerle et al. (2015). For gathering, transmission, and distribution pipelines, we use the EPA's GHGI emission factors of 0.31, 0.74, and 0.5 metric tons per year per mile, respectively (GHGI, 2021). For abandoned wells, we apply the measurement-based methane distributions for plugged and

unplugged wells as modelled by Williams et al. (2021). We estimate methane emissions due to well completions and workovers based on the EPA Greenhouse Gas Reporting Program (GHGRP) data for the Permian region (GHGPR, 2022), accounting for hydraulically fractured and non-hydraulically fractured well completions and workovers. Finally, we follow the procedure in Elvidge et al. (2015) to estimate a total of 106 bcf of gas flared in the Permian Basin in 2021, based on VIIRS detections. For each flaring detection at a facility or cluster of facilities, we assume an

average methane content of 80% in flared gas and a methane combustion efficiency of 91% (Plant et al. 2022).

**Table 4.** Summary of activity data and facility-scale methane emissions data sources or models

| Oil and gas methane source or facility category | Activity data (2021) | Facility-scale methane emission data, models and emission factors |
| --- | --- | --- |
| Oil and gas production well sites | 131,271 well sites | Methane emission models for low- and non-low producing well sites based on over 900 previous site-level measurement data (see Main Text) |
| Gathering and boosting compressor stations | 837 stations; 96,000 pipeline miles | Zimmerle et al. (2020); EPA GHGI |
| Gas processing facilities | 163 facilities | Mitchell et al. (2015), Marchese et al. (2015), Alvarez et al. (2018), EPA GHGI |
| Flaring related emissions | 1,560 detections | Elvidge et al. (2016); Lyon et al. (2021) |
| Transmission and distribution | 30 compressor stations; 44,480 pipeline miles | Subramanian et al. (2015); Zimmerle et al. (2015); Alvarez et al. (2018), Weller et al. (2020) |
| Abandoned wells | 218,155 wells | Williams et al. (2021); EPA GHGI |
| Well completions and workovers | 4,797 wells | EPA Greenhouse Gas Reporting Program (GHGRP) |


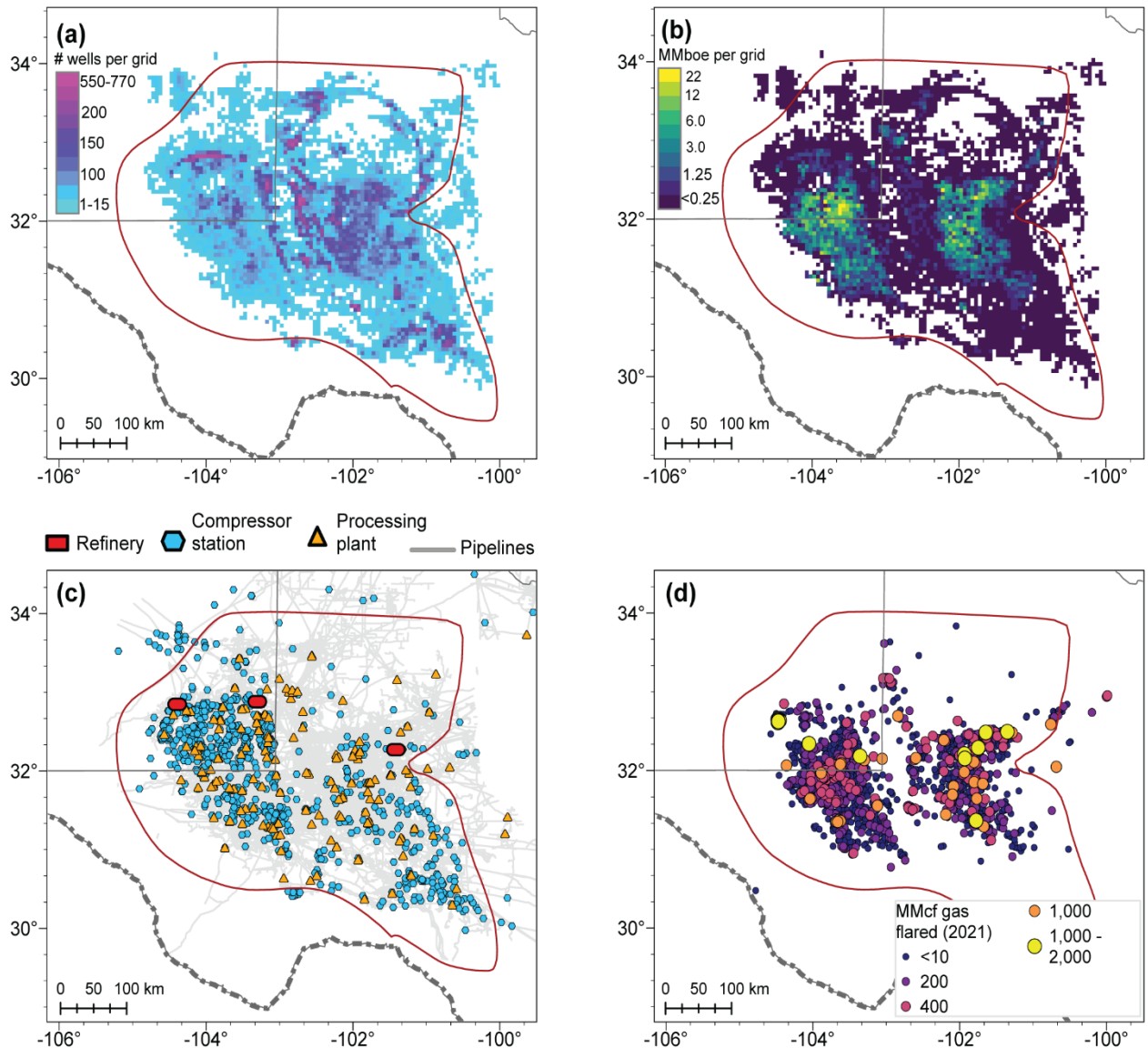

**Figure 13.** Permian Basin oil and gas activity data (Permian Basin boundary shown in red). **(a).** Density of oil and gas wells (5 km x 5 km grids). **(b).** Density of oil and gas production (MMboe, 5 km x 5 km grids), **(c).** Locations of major oil and gas infrastructure. **(d).** VIIRS-derived gas flaring detections. Data sources: OGIM_v1, Enverus (2021).

Our estimate for total Permian methane emissions is 3.1 (95% CI: 2.6—3.6) Tg in 2021. The uncertainty on our estimates reflects uncertainties in the mean facility-level methane emissions distributions as well as uncertainties in activity data. Based on the total methane production of 100 Tg, and assuming 80% methane content in produced natural gas, we estimate the Permian methane emissions represent a mean methane loss rate of 3.1% (95% CI: 2.6 – 3.6%) in 2021. Our estimate of total oil and gas Permian methane emissions, leveraging measurement-based methane emissions data, is approximately a factor of 3x higher than estimates from the EPA's gridded methane emission inventory (Zhang et al. 2020, Maasakkers et al. 2016, Shen et al. 2022), and ~16% higher than our previous estimate of ~2.7 Tg using 2018 activity data (Zhang et al. 2020), suggesting increasing methane emissions due to increasing

oil and gas activity in the intervening years (e.g., new oil and gas development and natural gas flaring related emissions).

Our bottom-up methane emissions estimate for each of the major oil and gas sectors in the Permian are based on measurements collected using facility-scale, ground-based measurement approaches, such as the EPA Other Test Methods (OTM-33A, e.g., Robertson et al. 2020) and dual tracer flux measurements (Mitchell et al. 2015). Additional

facility-scale emissions datasets include measurements collected using point source aerial measurement platforms (e.g., Cusworth et al. 2021, 2022), with higher minimum detection limits (e.g., ~10-20 kg/h; Cusworth et al. 2021) and detections of low-probability and intermittent high-magnitude emissions events.  Further studies are needed to develop statistically robust methods for integrating facility-scale ground-based datasets with such "top-down" datasets. Because of the paucity of facility-scale measurements for gathering and transmission pipelines, our use of

the EPA Greenhouse Gas Inventory methane emission factors may represent a low-bound on total estimated emissions for these sectors, as recent studies suggest the EPA emission factors could be significantly biased low (Yu et al. (2021)).

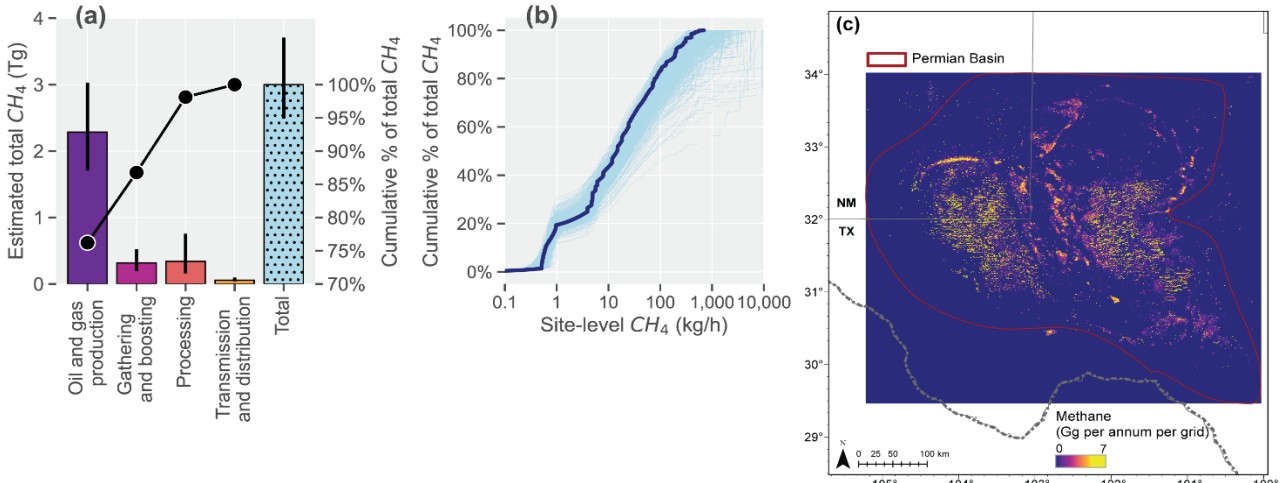

**Figure 14.** Bottom-up estimates of Permian Basin oil and gas methane emissions, based on measurement data. **(a).** Estimated total methane emissions for major oil and gas methane sources. The circle symbols (right y-axis) show the cumulative percentages of total methane emissions, indicating approximately 77% of estimated total methane emissions arise from oil and gas production activities, including methane slip from inefficient natural gas flaring. Error bars show the 95% confidence intervals on the mean total estimates for each sector, reflecting model uncertainty in

the mean distribution of site-level methane emissions and activity data **(b).** Cumulative percentages of total methane emissions as functions of modelled site-level methane emission rates for all facilities shown in a. The mean distribution for all sites is shown in solid dark-blue line while the light-blue lines show the uncertainty in the distributions obtained from 500 Monte-Carlo realizations of each site's modeled methane emissions. **(c).** High-resolution (~1 km x 1 km) spatial distribution of oil and gas methane emissions in the Permian Basin, based on facility-scale measurements.


Our high-resolution bottom-up methane inventory provides a first order estimate of the predominant sources of basin-level oil and gas methane emissions, as well as detailed spatial distribution of emissions, which supports further methane source attribution at the regional or basin-level, beyond attribution to individual high emitting point sources.

For example, our work suggests that oil and gas production facilities are the predominant methane sources in the Permian Basin, accounting for about 80% of total emissions (Figure 14). Our analysis shows clear methane hotspots concentrated in the Delaware (western half of the basin) and Midland (eastern half of the Permian, Figure 14) sub-basins, that closely aligns with the density of infrastructure, flaring, and production (Figure 13b, c, d). Improved bottom-up methane inventories also allow for the assessment of the distribution of facility-scale methane emissions,

revealing the relative contributions of both the high-emitting and low-emitting sources (Figure 14b). For example, in Figure 14b, we estimate that roughly 90% of the mean total methane emissions in the Permian Basin arise from sources that individually emit less than 100 kg/h/site, based on 2021 oil and gas activity, underscoring the importance of a large number of relatively low-emitting facilities in this region accounting for the vast majority of total Permian Basin-wide methane emissions.


## 5.    Data availability

OGIM_v1 can be accessed at https://doi.org/10.5281/zenodo.7466757 (Omara et al. 2022) in an open-access GeoPackage file format. OGIM_v1 was developed and tested using open-access software (Python 3.7 and QGIS). The current version of the publicly available OGIM database does not include compressor station locations for Russia

(shown in the map on Figure 6). Future updates to the OGIM database may include these datasets when appropriate permissions to make them publicly accessible are obtained. The updated bottom-up oil and gas methane emission inventory for the Permian Basin (for 2021) is available at https://doi.org/10.5281/zenodo.7466607 (Omara and Gautam, 2022) as a netcdf file.

## 6.    Code availability

Python 3.7 code used for database integration and visualization is available upon reasonable request.

## 7. Conclusions

       Advances in satellite methane remote sensing hold the promise of rapidly detecting and quantifying global

oil and gas methane emissions across multiple spatial scales, from area-aggregate sources to facility-scale assessment. However, effective characterization of remotely sensed oil and gas methane emissions in support of mitigation of avoidable emissions requires a comprehensive global geolocated oil and gas infrastructure inventory with detailed facility attributes. Such a comprehensive, granular, and global-in-context infrastructure database is also needed for developing and updating bottom-up oil and gas methane emissions inventories which are used as *a priori* data for

Bayesian inverse analysis of satellite observations for quantifying and attributing methane emissions. This work focuses on public-domain oil and gas datasets for all major facility categories that are significant methane emitters in order to develop a spatially explicit global database of oil and gas infrastructure. We acquired approximately six million features representing locational-based information for major oil and gas infrastructure categories, including oil and gas wells, natural gas compressor stations, gathering and processing facilities, LNG facilities, refineries,

storage facilities, oil and gas production data, and transportation pipelines. We further present an updated framework to develop improvements to bottom-up emission inventories using our infrastructure database, with inputs from other

multi-scale empirical data and modelling, to demonstrate a high-resolution emissions inventory for the entire Permian Basin which accounts for over 40% U.S. annual oil production. In addition, we show various examples of the applications of this database, including (i) tracking temporal changes in oil and gas activity in specific oil and gas producing basins, (ii) supporting the development of policy-relevant analytics and insights for effective methane mitigation, and (iii) enabling methane source attribution at the facility-scale and at regional scale. We finally provide an assessment of data gaps in the current version of the database, given our focus on acquisition and integration of public-domain datasets, which can be limited in certain oil and gas producing countries especially in Asia and Africa. Further efforts are needed to help fill in these data gaps, particularly the gaps in the locational information of major oil and gas facilities and gaps in the availability of relevant facility attributes. Such efforts could include the development of deep-learning methods for automatically identifying and classifying oil and gas features in high-resolution satellite imagery. The OGIM database, which we anticipate updating on a regular cadence, (that is, at least once annually) as new datasets become available, fills a crucial oil and gas geospatial data need, in support of the assessment, attribution, and mitigation of global oil and gas methane emissions.

### Author contributions

MO and RG conceptualized the study. OGIM data acquisition, formal analysis, interpretation, and visualization were performed by MO, RG, MAO, and AH, with assistance from AF, KM, and GH. MethaneAIR $xCH_4$ plume data were collected and processed by AP, JF, CCM, and SW. MO wrote the manuscript with contributions from all co-authors.

### Competing interests

The authors declare that they have no conflict of interest.

### Acknowledgments

We are grateful to Benjamin Hmiel, Jack Warren, Scott Seymour, David Calhoun, and Emma Bishop for their review of an earlier version of the OGIM database. We thank Christopher Elvidge and the Earth Observation Group (at Colorado School of Mines) for permission to include the 2021 VIIRS flaring data (Elvidge et al. 2015) in the OGIM database.

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

# Appendix A: OGIM database schema

| Table A1: Attributes present in all layers | | | | | |
|---|---|---|---|---|---|
| Attribute name | Data type | Allow nulls? | Description | Valid values and/or example values | Notes |
| CATEGORY | string | no | Category of oil and gas infrastructure to which the record belongs | e.g., 'OIL AND NATURAL GAS WELLS', 'NATURAL GAS COMPRESSOR STATIONS', 'OIL AND NATURAL GAS PIPELINES' | Within a GeoPackage layer, all values for CATEGORY are the same (and the CATEGORY value is identical to the layer name). |
| COUNTRY | string | no | Country in which the record resides. Where possible, country name matches the name in the UN Member State list. | e.g., 'GERMANY'; 'AFGHANISTAN, TURKMENISTAN' | LineString and Polygon features may fall in 2+ countries; in these cases, COUNTRY field contains a comma-separated list of these countries in alphabetical order |
| OGIM_ID | integer | no | Unique identifier for each record in the Geopackage. Values do not "reset" for each layer. | | |
| ON_OFFSHORE | string | no | Indicates whether the asset lies onshore, offshore, or both. | Valid values: 'ONSHORE'; 'OFFSHORE'; 'ONSHORE, OFFSHORE' | Only LineString and Polygon features may fall both on and offshore, so only these geometries may have the value 'ONSHORE, OFFSHORE' |
| REGION | string | no | World region in which the record lies. When possible, region aligns with the IEA's Energy Region classifications. | valid: 'AFRICA', 'ASIA PACIFIC', 'CENTRAL AND SOUTH AMERICA', 'EURASIA', 'EUROPE', 'NORTH AMERICA' | |
| SRC_DATE | string | no | Date on which the record's original source was published. | ex: '2014-06-01' | |
| SRC_REF_ID | string | no | ID number(s) linking the record to its corresponding source in the "Data_Catalog" table. | ex: '22'; '89, 92' | There are some records that list two SRC_REF_IDs separated by a comma in this column if that infrastructure category was derived from multiple data sources; for example, "89, 92". |
| STATE_PROV | string | yes | State or province in which the record resides. | ex: 'TEXAS'; 'ALBERTA' | |

| Attribute name | Data type | Allow nulls? | Description | Valid values and/or example values | Notes |
|---|---|---|---|---|---|
| **geometry** | geometry | no | Vertices of the feature's geometry. Formatted as well-known text (WKT) representations of the geometries. | ex: 'POINT (67.42377999999999 37.21161)' | |

**Table A2: Attributes present in all oil and natural gas wells and other major oil and gas infrastructure layers**

| Attribute name | Data type | Allow nulls? | Description | Valid values and/or example values | Notes |
|---|---|---|---|---|---|
| **FAC_ID** | string | yes | Unique ID used by the original source agency. | ex: 'BGBR0230'; '126162'; '5609/10-01' | |
| **FAC_NAME** | string | yes | Name of the infrastructure asset. | | |
| **FAC_STATUS** | string | yes | Operational status of the infrastructure asset, according to the original source. | ex: 'ACTIVE'; 'SUSPENDED'; 'TEMPORARILY CLOSED' | FAC_STATUS of "N/A" means facility status information not reported or available in the original dataset. |
| **FAC_TYPE** | string | yes | Detailed information on type of facility. | ex: "EXPORT FACILITY"; "NGL FRACTIONATION FACILITY" | |
| **INSTL_DATE \*** | date | yes | Date the facility or asset was installed, in YYYY-MM-DD format. | ex: 1994-02-17 | Some data sources only included an installation year, or a month-year combo. We fill these values with their month or date values as '01'. For example, if installation was reported only as 2012, we standardize this value in the INSTL_DATE attribute as '2012-01-01' |
| **LATITUDE** | float | no | Latitude of Point features (decimal degrees, WGS 1984) | ex: 30.11438 | |
| **LONGITUDE** | float | no | Longitude of Point features (decimal degrees, WGS 1984) | ex: -93.29659 | |
| **OGIM_STATUS** | string | yes | Standardized version of FAC_STATUS, created by OGIM team to sort and simplify statuses reported by the original source. | valid: 'PERMITTING'; 'UNDER CONSTRUCTION'; 'OPERATIONAL'; 'PROPOSED'; 'DRILLING'; 'COMPLETED'; 'PRODUCING'; 'INACTIVE'; 'ABANDONED'; 'INJECTING'; 'STORAGE, MAINTENACE, OR OBSERVATION'; 'OTHER' | |

| OPERATOR | string | yes | Name of the infrastructure's operator, according to the orginal source. | EX: 'YSUR ENERGÍA ARGENTINA S.R.L.'; 'PETROBRAS'; 'DCP MIDSTREAM, LP' | No modifications have been made to standardize operator names or associate subsidiaries with parent companies. |

\* = attribute not present for wells

**Table A3: Attributes present in the oil and natural gas wells layer only**

| Attribute name | Data type | Allow nulls? | Description | Valid values and/or example values | Notes |
|---|---|---|---|---|---|
| COMP_DATE | string | yes | Date well construction was completed. | ex: 2019-12-13 | |
| DRILL_TYPE | string | yes | Drilling direction of the well. | ex: 'HORIZONTAL'; 'VERTICAL'; 'DIRECTIONAL'; | Conventional' indicates a vertical well; 'Unconventional' indicates a horizontal or 'hydrofracking' well |
| SPUD_DATE | string | yes | Date well was first spudded | ex: 2019-04-11 | |

**Table A4: Attributes present in the oil and natural gas pipelines layer only**

| Attribute name | Data type | Allow nulls? | Description | Valid values and/or example values | Notes |
|---|---|---|---|---|---|
| PIPE_DIAMETER_MM | float | yes | Pipe diameter in millimeters. | ex: 88, 114 | |
| PIPE_LENGTH_KM | float | no | Length of pipeline segment in kilometers. | ex: 4.45; 90.4; 1130 | Pipeline length is calculated and standardized (in units of km) for each feature in the OGIM data, even if the original data source reports a length value. |
| PIPE_MATERIAL | float | yes | Material pipeline is made of. | ex: 'STEEL'; 'POLYETHYLENE'; 'CARBON STEEL 5L GRADE X 65' | |

**Table A5: Attributes present in basins, fields, and license blocks layers only**

| Attribute name | Data type | Allow nulls? | Description | Valid values and/or example values | Notes |
|---|---|---|---|---|---|
| NAME | string | yes | The name of the basin, field, or license block | ex: 'PERMIAN' | |
| AREA_KM2 | float | no | Area of polygon in sq. kilometers. | ex: 37200; 186000 | Area is calculated and standardized (in units of $km^2$) for each feature in the OGIM data, even if the original data source reports a length value. |
| RESERVOIR_TYPE | string | yes | Hydrocarbon(s) produced by the reservoir, or the phase of production the reservoir is in. | ex: 'OIL'; 'OIL AND GAS'; 'CONDENSATE'; 'EXPLORATION AND EXPLOITATION'; 'COALBED METHANE' | |

| Table A6: Attributes present in flaring detections only | | | | | |
|---|---|---|---|---|---|
| **Attribute name** | **Data type** | **Allow nulls ?** | **Description** | **Valid values and/or example values** | **Notes** |
| **AVERAGE_FLARE_T EMP _K** | integer | no | Average flare temperature in Kelvin. | ex: 1020, 2119 | |
| **DAYS_CLEAR_OBSE RVA TIONS** | integer | no | Number of clear days for which flares were detected. | ex: 123, 381 | |
| **FLARE_YEAR** | integer | no | Year for which detections occurred. | valid: 2021 | |
| **GAS_FLARED_MMC F** | float | no | Estimated volume of gas flared in million cubic feet per year. | | |
| **SEGMENT_TYPE** | string | yes | Oil and gas industry segment to which the flaring detection belongs. | valid: 'GAS DOWNSTREAM'; 'OIL DOWNSTREAM'; 'UPSTREAM' | |

**Table A7: Additional attributes in infrastructure layers**

| Attribute name | Data type | Allow nulls? | Description | In which layer(s) is attribute present? |
|---|---|---|---|---|
| OIL_BBL | float | yes | Oil production in barrels per year | PR |
| GAS_MCF | float | yes | Gas production in thousands of cubic feet per year | PR |
| WATER_BBL | float | yes | Water production in barrels per year | PR |
| CONDENSATE_BBL | float | yes | Condensate production in barrels per year | PR |
| GAS_CAPACITY_MMCFD | float | yes | Facility capacity for natural gas, in million cubic ft. per day. | CS, GP, ID, LNG, PL, TM |
| GAS_THROUGHPUT_MMCFD | float | yes | Facility throughput for natural gas, in million cubic ft. per day. | CS, GP, ID, LNG, PL, TM |
| LIQ_CAPACITY_BPD | float | yes | Facility capacity for O&G liquids, in barrels per day. | CS, GP, ID, LNG, PL, R, TM |
| LIQ_THROUGHPUT_BPD | float | yes | Facility throughput for O&G liquids, in barrels per day. | CS, GP, ID, LNG, PF, PL, R, TM |
| NUM_STORAGE_TANKS | integer | yes | Number of storage tanks at the facility. | CS, GP, ID, LNG, PF, R, TM |
| NUM_COMPR_UNITS | integer | yes | Number of compressor units present at facility. | CS, GP, TM |
| SITE_HP | float | yes | Horsepower of the facility. | CS, GP |
| COMMODITY | string | yes | Hydrocarbon(s) contained in the infrastructure | PL, TM |

**Key for Table H:**
CS = Compressor Stations
GP = Gathering and Processing
ID = Injection and Disposal
LNG = Liquified Natural Gas Facilities

PL = Pipelines
PR = Production well sites
R = Refineries
TM = Petroleum Terminals