# Peer review of "Developing a spatially explicit global oil and gas infrastructure database for characterizing methane emission sources at high resolution"

_Earth System Science Data, 2022_

## Author Comment (AC1)

**Response to reviewer comments on:**

**Omara, M., Gautam, R., et al. Developing a spatially explicit global oil and gas infrastructure database for characterizing methane emission sources at high resolution (https://essd.copernicus.org/preprints/essd-2022-452/)**

**Reviewer 1**

This is a great and valuable dataset, and I appreciate the authors hard work in aggregating and making this data publicly available. I downloaded the dataset and it was easy to navigate, visualize, and extract layers. The methods and uncertainties are well described in the paper. I have a few comments listed below:

We thank the reviewer for these helpful feedback on our manuscript.

-The gap assessment and spatial assessment in this study are both very useful for error quantification. However, I think there is an additional error term or test that could be helpful especially in context of your MethaneAir analysis - the gaps that exist in a "dense" country. Or another way, "if MethaneAir observes plume in the Permian, how close is the nearest piece of GIS infrastructure, and is it right to attribute it to that plume?"  You show proof of concept with MethaneAir, but EDF has gathered extensive and independent facility-scale observations with attribution through PermianMap where you could do this test rigorously.

We appreciate this interesting comment from the reviewer regarding methane plume source attribution assessments in regions with high density of oil and gas infrastructure.

The assessment of the proximity of a remotely sensed methane plume to an oil and gas methane emitting facility is dependent on several factors, including the methane plume characteristics (e.g., plume shape and spatial extent), the spatial resolution and georectification accuracy of the methane remote sensing data, and the spatial accuracy of the database being used for methane source attribution.

While we provide a general estimate of the spatial accuracy of the Oil and Gas Infrastructure Mapping (OGIM) data, the detailed assessment of methane plume proximity to infrastructure points and whether it is "right to attribute" the plume to a given location, as suggested by the reviewer, is outside the scope of this study. We note that the *"is it right to attribute"* question is also dependent on the analyst's ability to correctly characterize the remotely sensed methane plume as a true positive detection, and to correctly classify it as a methane plume attributable to an oil and gas source (and not other methane sources, such as wetlands).

We believe our spatial accuracy and gap assessment, as presented in the manuscript, is sufficiently robust in demonstrating the comprehensiveness of the OGIM spatial data coverage (and data gaps therein) as well as the spatial accuracy of point locations of key oil and gas infrastructure types. In addition, we show various use cases/analytics and insights that can be derived using the OGIM data and note that these are not intended as an exhaustive and detailed characterization of the applications of the database, which can vary by user.

-Related to previous comment: in addition to randomly sampling GIS infrastructure and comparing to visible imagery, why not assemble an independent list of GIS infrastructure solely through visible inspection (say 100 of so in various basins/countries) and then see if your GIS database has elements near your visual list? This would be a more blinded test of gaps in the inventory and spatial accuracy.

We thank the reviewer for this suggestion and acknowledge that such blinded assessments are useful in further assessing the OGIM data coverage and spatial accuracy.

To help provide further assessment of the OGIM data coverage and spatial accuracy, we use a machine-learning derived dataset of oil and gas infrastructure in the Permian Basin, developed by training machine learning (ML) models to automatically detect and classify locations of oil and gas infrastructure in AirBus SPOT imagery (1.5 m pixel resolution) for 2019. Further details of the model development and the ML-derived dataset can be found in Lyon et al. (2020). The original ML-dataset included over 190,000 locations in the Permian, which we filter to 35,107 locations with reported model raw confidence of >95%, indicating high confidence in the likelihood of the model detection being an oil and gas facility. The ML-derived locations are reported for the centroid of the facility footprint based on the satellite imagery. We note that even with this filter of high-confidence detections, it is possible that there are a small undetermined fraction of non-oil and gas infrastructure locations with footprints that are similar in appearance to oil and gas infrastructure footprints in satellite imagery. Nevertheless, this ML-derived dataset represents a very large and independent dataset of oil and gas infrastructure locations in the Permian Basin and provides a unique opportunity for a comprehensive comparison with the OGIM data to further assess OGIM data coverage and spatial accuracy.

For each of the 35,107 high-confidence ML-derived locations, we use a $k$-dimensional binary tree algorithm to search for its nearest neighbor in the OGIM dataset and compute the distance (in meters) between the ML-derived location and the OGIM nearest neighbor. Setting a distance threshold of 100-m as an approximate dimension (length/width) of the typical oil and gas facility in the Permian, we find 33,620 OGIM locations are within 100-m of the ML-derived locations, suggesting comprehensive coverage (96%) with high spatial accuracy at the set distance threshold. This coverage increases to 97% at 250-m threshold and 99% at 500-m threshold. We note that the Permian is a highly dynamic basin in terms of oil and gas activity, and had ~300 new well pad development per month in 2019 (Lyon et al. 2020). Additionally, public data reporting in this region can have reporting lags of more than three to six months. Thus, while it is possible to track monthly trends in new oil and gas development using machine learning approaches (depending on satellite imagery refresh rate), the public data reporting lag and update frequencies could help explain the <100% coverage assessed herein. A histogram of the computed distances between the ML-derived locations and the OGIM locations is shown in the figure below.

The above assessment provides an independent check on the OGIM data coverage and spatial accuracy in a dynamic oil and gas basin with dense oil and gas infrastructure. Future data verification work should leverage similar approaches to further characterize data coverage and spatial accuracy in other regions, as it is infeasible for us to manually develop, on a scale

suggested by the reviewer, an independent verification dataset based on manual detection and classification of oil and gas infrastructure footprints in satellite imagery.

We include the above assessment as supplementary information to our manuscript.

**(a)**

[Figure]

[Figure]

**Figure B1**. (a) Visualization of a machine-learning derived dataset of point locations of oil and gas infrastructure in the Permian Basin (Lyon et al. 2020) and the OGIM dataset in the same region. The red points show the filtered high-confidence ML-derived dataset, the red polygons show a 50-m radius buffer around each ML-derived point, and the cyan triangles show the OGIM oil and gas infrastructure points. (b) Histogram of the distance between an ML-derived dataset and its nearest neighbor in the OGIM dataset. The dashed dark-red lines show the distance thresholds of 100-m, 250-m, and 500-m.

-The analysis of Permian emissions based on the inventory raises confusion. After application of literature-based emission factors, you get an emission budget/basin loss rate on par with previous atmospheric inversion studies (3.1 Tg). However, the conclusion that this budget is dominated by upstream low-emitting sources is a very different conclusion than what's been found in previous studies, some of which have been published by EDF. For example, from the abstract of Lyon et al. (2021) [https://doi.org/10.5194/acp-21-6605-2021]: "the Permian Basin is in a state of overcapacity in which rapidly growing associated gas production exceeds midstream capacity and leads to high methane emissions." Also, in your description of activity/emission factors, you use EPA GHGI emission factors for gathering pipelines - however from Yu et al. (2021) [https://doi.org/10.1021/acs.estlett.2c00380]: " In this study, we use methane emission measurements collected from four recent aerial campaigns in the Permian Basin, the most prolific O&G basin in the United States, to estimate a methane emission factor for gathering lines. From each campaign, we calculate an emission factor between 2.7 (+1.9/–1.8, 95% confidence interval) and 10.0 (+6.4/–6.2) Mg of CH4 year–1 km–1, 14–52 times higher than the U.S. Environmental Protection Agency's national estimate for gathering lines and 4–13 times higher than the highest estimate derived from a published ground-based survey of gathering lines." Application these alternative emission factors which have been observed in the Permian would certainly change your conclusion about upstream/midstream. Without this context explicitly stated in your Permian analysis, this section reads as an attempt to arrive at a certain prescribed conclusion, which I don't believe is your intent. The preferred approach would be to make an ensemble of estimates based on the many emission factor distributions that have

been recently observed. Or at minimum, the manuscript should state that upstream/diffuse result from your inventory is not totally consistent with other studies from the Permian. However, this seems outside the scope of a data description paper and may be worth total removal/saving for another analysis.

We appreciate the reviewer's helpful comments on the section of our manuscript regarding methane emissions inventory in the Permian Basin.

We note that the Lyon et al. (2020) study does not claim that emissions in the Permian are dominated by the midstream sector, and the study does not provide an estimate of the proportion of the total measured emissions attributable to the midstream sector. As the reviewer noted, the study suggested that "insufficient capacity of midstream infrastructure for handling and delivering rapidly growing rates of natural gas" contributed to the high observed emission rates pre-COVID. Insufficient capacity of midstream infrastructure (pipelines, compressor stations, processing plants) could create production takeaway bottlenecks both at or upstream (i.e., at oil and gas production sites) of the midstream facilities, as further evidenced by the reported high flaring and flare malfunctions (i.e., venting) of produced associated gas at the time (Lyon et al. (2020). We note that our estimate is also for the 2021 post-COVID era, when Permian takeaway capacity had considerably improved (Varon et al. 2023).

Our use of the EPA methane emissions factors for the gathering pipelines reflects the paucity of facility-scale ground-based measurement datasets for methane emissions from gathering pipelines. For consistency, we only used available data from previous facility-scale ground-based measurements in our emissions estimates for all oil and gas sectors (supplementing with the EPA Greenhouse Gas Inventory emission factors, where such measurements were not available). The high minimum detection limits of most of today's point-source "top-down" aerial measurements, and the highly intermittent and often short-duration nature of the high-magnitude emissions typically observed by these platforms, require further studies on statistically robust methods for integrating such datasets with the facility-scale ground-based measurements. We include the following sentences in the revised manuscript:

- *"For consistency, our bottom-up methane emissions estimate for each of the major oil and gas sectors are based on measurements collected using facility-scale, ground-based measurement approaches, such as the EPA Other Test Methods (OTM-33A, e.g., Robertson et al. 2020) and dual tracer flux measurements (Mitchell et al. 2015). Additional facility-scale emissions datasets include measurements collected using point source aerial measurement platforms (e.g., Cusworth et al. 2021, 2022), with higher minimum detection limits (e.g., >10-20 kg/h; Cusworth et al. 2021) and detections of low-probability and intermittent high-magnitude emissions events. Further studies are needed to develop statistically robust methods for integrating facility-scale ground-based datasets with such "top-down" datasets. Because of the paucity of facility-scale ground-based measurements for gathering and transmission pipelines, our use of the EPA Greenhouse Gas Inventory methane emission factors may represent a low-bound on total estimated emissions for these sectors, as recent studies suggest the EPA emission factors could be biased low (Yu et al. (2021)."*

In addition, we provide 95% confidence bounds on our total methane emissions estimates of +24%/-17% or 2.6—3.6 Tg, resulting from uncertainty in the mean facility-level emissions distributions and uncertainty in oil and gas activity data, particularly for gathering natural gas compressor stations.

To the best of our knowledge, there have not been any previous comprehensive "top-down" measurement-based studies characterizing total methane emissions by oil and gas sector in the Permian, i.e., apportionment of total methane emissions to specific facility categories such as well pads and compressor stations. Our Permian inventory, in addition to demonstrating one of the key applications of the OGIM database, provides an improved "bottom-up" estimate and methane source allocation based on available facility-level measurement datasets.

-On the Zenodo DOI webpage it states that datasets for Russian compressors and VIIRS are not included in the dataset due to permissions. I did not see that description also written in the manuscript, where it should also be.

We have updated the current version of the OGIM database to include the VIIRS flaring dataset for the year 2021. We have included the following description in the Data Availability section of the manuscript:

- *"The current version of the publicly available OGIM database does not include compressor station locations for Russia (shown in the map on Figure 6). Future updates to the OGIM database may include these datasets when appropriate permissions to make them publicly accessible are obtained."*

**References:**

Lyon, D. R., Hmiel, B., Gautam, R., Omara, M., Roberts, K. A., Barkley, Z. R., Davis, K. J., Miles, N. L., Monteiro, V. C., Richardson, S. J., Conley, S., Smith, M. L., Jacob, D. J., Shen, L., Varon, D. J., Deng, A., Rudelis, X., Sharma, N., Story, K. T., Brandt, A. R., Kang, M., Kort, E. A., Marchese, A. J., and Hamburg, S. P.: Concurrent variation in oil and gas methane emissions and oil price during the COVID-19 pandemic, Atmos. Chem. Phys., 21, 6605–6626, https://doi.org/10.5194/acp-21-6605-2021, 2021.

Varon, D. J., Jacob, D. J., Hmiel, B., Gautam, R., Lyon, D. R., Omara, M., Sulprizio, M., Shen, L., Pendergrass, D., Nesser, H., Qu, Z., Barkley, Z. R., Miles, N. L., Richardson, S. J., Davis, K. J., Pandey, S., Lu, X., Lorente, A., Borsdorff, T., Maasakkers, J. D., and Aben, I.: Continuous weekly monitoring of methane emissions from the Permian Basin by inversion of TROPOMI satellite observations, Atmos. Chem. Phys. Discuss. [preprint], https://doi.org/10.5194/acp-2022-749, in review, 2022.

Yu, J., Hmiel, B., Lyon, D.L., Warren, J., Cusworth, D.H., Duren, R.M., Chen, Y., Murphy, E.C., Brandt, A.R. Methane Emissions from Natural Gas Gathering Pipelines in the Permian Basin, Env. Sci. Technol., 9, 969-974, 2022, https://doi.org/10.1021/acs.estlett.2c0038

---

## Author Comment (AC2)

**Response to reviewer comments on:**

**Omara, M., Gautam, R., et al. Developing a spatially explicit global oil and gas infrastructure database for characterizing methane emission sources at high resolution (https://essd.copernicus.org/preprints/essd-2022-452/)**

**Reviewer 2**

The oil and gas infrastructure mapping is very important to monitoring and modeling the GHG emissions for limiting climate change. This work provides the most complete dataset so far that collects infrastructure information from online sources. The infrastructure type and geolocation from this work can be very useful to the GHG emission inventory developments and modeling from facility to regional and global scale with remote sensing images. Also, it can also be used as the ground-truth dataset for the oil and gas infrastructure identification with remote sensing images and machine learning approach. The methods are clearly described, and the paper also provides a detailed bottom-up emission inventory case study using this dataset.

We thank Reviewer 2 for these helpful feedback on our manuscript.

I'm very excited to download and look at the OGIM_v1 dataset (OGIM_v1.gpkg), however, I found the currently provided dataset should be further modified before it can be used in other research:

1.      It would be better to provide the mapping from column names as well as the shortcuts used in the entries (especially for the "FAC_TYPE") to their detailed meanings, maybe provide a table in the supplementary or in the description of Zenodo?

We have included an appendix to the Main Text with full description of the OGIM data attributes.

2.      Figure 6. I see oil and natural gas infrastructures in some countries, such as China and India, but I cannot find them in the OGIM_v1 dataset. There is no "China" or "Indian" in the "COUNTRY".

There is indeed oil and gas infrastructure datasets for China and India in the OGIM database. We note that these are countries with limited or no public data for oil and natural gas wells (Figure 6 (a)) and, as such, searching the OGIM database for wells information may return null values. However, also as shown in Figure 6, the OGIM database includes several other publicly available oil and gas infrastructure datasets for these countries, including LNG facilities, pipelines, refineries, and offshore platforms.

3.      It will be interesting to see how the emissions of the natural gas compressor as well as the natural gas flaring changed from remote sensing images after the Ukraine war if the infrastructure information is provided by the dataset. So, if the Russian data

are not included in the current version, it should be stated in the paper unless it will be available soon.

We have included the following description in the Data Availability section of the manuscript:

- *"The current version of the publicly available OGIM database does not include compressor station locations for Russia (shown in the map on Figure 6). Future updates to the OGIM database may include these datasets when appropriate permissions to make them publicly accessible are obtained."*

4.      I tried to extract the pipeline from the OGIM_v1 dataset, however, there is no such type either from "FAC_TYPE" or "geometry". The same issue also exists for the fields or basins.

The OGIM_v1 dataset does indeed include pipeline datasets and oil and gas fields and basins. The following are the layer names in the OGIM dataset:

```
['Oil_and_Natural_Gas_Wells',
 'Natural_Gas_Compressor_Stations',
 'Gathering_and_Processing',
 'Tank_Battery',
 'Offshore_Platforms',
 'LNG_Facilities',
 'Crude_Oil_Refineries',
 'Petroleum_Terminals',
 'Injection_Disposal_and_Underground_Storage',
 'Stations_Other',
 'Natural_Gas_Flaring_Detections',
 'Equipment_and_Components',
 'Oil_and_Natural_Gas_Production',
 'Oil_Natural_Gas_Pipelines',
 'Oil_and_Natural_Gas_Fields',
 'Oil_Natural_Gas_Basins',
 'OGIM_v1_Data_Catalog'
]
```

We note that the OGIM_v1 dataset was developed and tested using open-source software, including Python 3.7 and QGIS. We have not tested and do not guarantee that the dataset will be accessible in other proprietary software or GIS programs. We have added the following sentence in the "Data Availability" section:

- *"OGIM_v1 was developed and tested using open-access software (Python 3.7 and QGIS)."*

5.      Line 360-365: "We quantitatively assess data quality in each country for which open oil and gas data for these facilities are available in the OGIM_v1 database". This will be another advantage of this dataset if all the entries are labeled with quality scores, and users with different research purposes can easily select them without any data cleaning processes. But I did not see the score from the dataset.

I uploaded a simple test code, maybe I missed them?

We have uploaded an update to the OGIM database (OGIM_v1.1) on Zenodo that includes the addition of VIIRS flaring data. Data quality scores, as described in the Main Text, are incorporated in the database attributes.